# Structural basis for shape-selective recognition and aminoacylation of a D-armless human mitochondrial tRNA

Bernhard Kuhle [1] ✉, Marscha Hirschi[2], Lili K. Doerfel [2], Gabriel C. Lander [2] & Paul Schimmel [1,3]

Human mitochondrial gene expression relies on the specific recognition and aminoacylation of mitochondrial tRNAs (mtRNAs) by nuclear-encoded mitochondrial aminoacyl-tRNA synthetases (mt-aaRSs). Despite their essential role in cellular energy homeostasis, strong mutation pressure and genetic drift have led to an unparalleled sequence erosion of animal mtRNAs. The structural and functional consequences of this erosion are not understood. Here, we present cryo-EM structures of the human mitochondrial seryl-tRNA synthetase (mSerRS) in complex with mtRNA$^{Ser(GCU)}$. These structures reveal a unique mechanism of substrate recognition and aminoacylation. The mtRNA$^{Ser(GCU)}$ is highly degenerated, having lost the entire D-arm, tertiary core, and stable L-shaped fold that define canonical tRNAs. Instead, mtRNA$^{Ser(GCU)}$ evolved unique structural innovations, including a radically altered T-arm topology that serves as critical identity determinant in an unusual shape-selective readout mechanism by mSerRS. Our results provide a molecular framework to understand the principles of mito-nuclear co-evolution and specialized mechanisms of tRNA recognition in mammalian mitochondrial gene expression.

Mitochondria are essential eukaryotic organelles, responsible for the maintenance of cellular bioenergetics. Resulting from their endosymbiotic origin[1], they retained a reduced genome (mtDNA) that encodes 13 essential subunits of the respiratory chain complexes, which are expressed by a highly divergent mitochondrial translation machinery[2]. The aminoacylation reaction is the first and fidelity-determining step of mitochondrial translation, in which each of 22 mtDNA-encoded tRNAs is charged with the cognate amino acid by one of 19 nuclear-encoded mitochondrial aminoacyl-tRNA synthetases[2,3]. The rules and accuracy of mitochondrial gene expression are thus established by 'chimeric' enzyme-RNA complexes of dual genetic origin, exposed to distinct evolutionary pressures and mutation rates.

The aaRS-tRNA interactions underlying aminoacylation are among the oldest in biology, conceivably dating back to the onset of cellular evolution[4]. Accordingly, the canonical tRNA structure, as adapter between mRNA codons and the amino acid sequence of proteins, is highly conserved throughout extant cellular life[5]. Usually consisting of 76 nucleotides, it is characterized by a four-armed cloverleaf-like secondary structure that arranges into an L-shaped three-dimensional fold held together by an intricate network of tertiary interactions[5–7]. Conserved between tRNAs from prokaryotes to the eukaryote cytoplasm, this fold provides a structurally homogenous platform for interactions with other components of the translation machinery, from processing enzymes to ribosomes[8,9]. Embedded into this common structural scaffold are unique sets of highly conserved nucleotides, termed 'identity elements', which ensure specific recognition and aminoacylation of canonical tRNAs by their cognate aaRSs[10,11].

[1]Department of Molecular Medicine, The Scripps Research Institute, La Jolla, CA 92037, USA. [2]Department of Integrative Structural and Computational Biology, The Scripps Research Institute, La Jolla, CA 92121, USA. [3]The Scripps Florida Research Institute at the University of Florida, Jupiter, FL 33458, USA. ✉ e-mail: bkuhle@scripps.edu

The mitochondrial tRNAs of bilaterian animals show an unparalleled divergence from this paradigm[3,12,13]. High mutation rates and genetic drift in the mtDNA genome eroded much of the structure and sequence information in mtRNAs that is essential for canonical tRNA function. The loss of identity elements, accumulation of weak base-pairs, and mismatches in stem regions, insertions, and even deletions of entire domains gave rise to bizarre and fragile mtRNAs of unique structural diversity[3,12–14]. This degeneration and resulting instability are a major underlying cause for the susceptibility of mtRNAs and mt-aaRSs to mutations that compromise mitochondrial translation and lead to a wide range of human diseases[12,15–17].

Despite their importance for mitochondrial function and human health, the molecular and functional properties underlying animal mitochondrial aaRS-tRNA interactions, in particular the mechanisms of selective mtRNA recognition and aminoacylation, are poorly understood, and it remains unclear how the long-term functional decline of mitochondrial gene expression is prevented, despite a continuous accumulation of mutations in mtRNA genes. Here, we provide insight into these longstanding questions by determining high-resolution structures of an animal mitochondrial aaRS-tRNA complex.

We reconstituted the complex between the human mitochondrial seryl-tRNA synthetase (mSerRS) and the D-armless mtRNA^Ser(GCU), the shortest and most divergent mammalian mtRNA (Supplementary Fig. 1)[12,18], and determined its structure using single-particle cryo-EM. Combined with X-ray crystallographic, mutational, and kinetic data, our results provide molecular level insights into the mechanisms underlying protein-tRNA recognition and mito-nuclear co-evolution in mammalian mitochondrial gene expression.

## Results

### Structures of free mSerRS and the mSerRS-mtRNA^Ser complex

To understand the molecular basis for tRNA-recognition in human mitochondria, we first determined the high-resolution crystal structure of human mSerRS bound to 5′-O-[N-(L-seryl)sulfamoyl] adenosine (SerSA), a non-hydrolysable analog of the seryl-adenylate intermediate (Fig. 1a, b; Supplementary Table 1). Human mSerRS, like other eukaryotic and bacterial SerRSs[19], is an $\alpha_2$ homodimer with dimerization mediated by the catalytic $\alpha + \beta$-core domain that defines class II synthetases[20]. Appended to each catalytic domain is an N-terminal domain, which forms a long coiled-coil 'helical arm' of two antiparallel α-helices that protrudes away from the synthetase core. Both active sites are occupied by a SerSA molecule (Fig. 1b and Supplementary Fig. 2a), bound similar to previously described SerRS structures[21–23]. Overall, human mSerRS is similar in sequence and structure to its bovine homolog[21], including short peptide extensions at both termini, the N-terminal 'N-helix' and C-terminal 'C-tail' (Fig. 1c and Supplementary Fig. 2b), which are conserved across mammalian mSerRSs but absent from non-mitochondrial homologs (Supplementary Fig. 3)[21].

To reconstitute the mSerRS-mtRNA^Ser complex, we incubated the recombinantly purified mSerRS with excess of SerSA ligand and in vitro-transcribed mtRNA^Ser(GCU). Using single-particle cryo-EM, we determined the structures of mSerRS-SerSA bound with either wild-

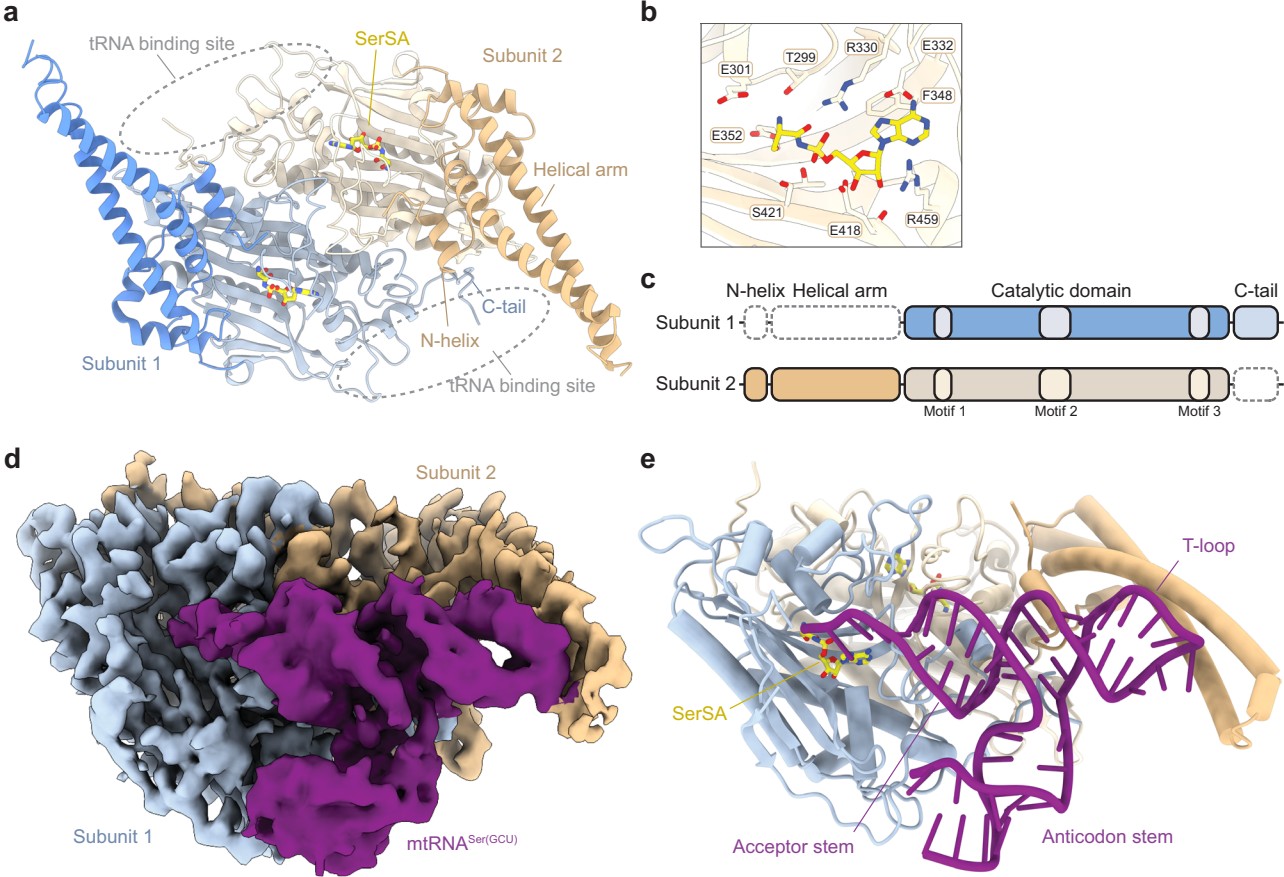

**Fig. 1 | Structures of human mSerRS bound to SerSA and mtRNA^Ser(GCU). a** Crystal structure of human mSerRS bound to SerSA (yellow sticks). mSerRS is shown in cartoon presentation in light blue (subunit 1) and wheat (subunit 2). Dashed circles indicate the two identical tRNA binding sites on opposite sides of the mSerRS homo dimer. **b** Binding pocket for SerSA in the mSerRS crystal structure. Interacting residues are shown as sticks. **c** Domain presentation of mSerRS. Dashed lines indicate structural domains not defined in subunits 1 (top) and 2 (bottom) in the cryo-EM reconstruction of mtRNA^Ser(GCU)-bound mSerRS shown in (**d**) and (**e**). **d** Segmented cryo-EM map of the mSerRS-mtRNA^Ser(GCU)-TL complex. **e** Structure of the mSerRS-mtRNA^Ser(GCU)-TL complex in cartoon presentation. mSerRS is colored as in (**a**), the tRNA is shown in purple.

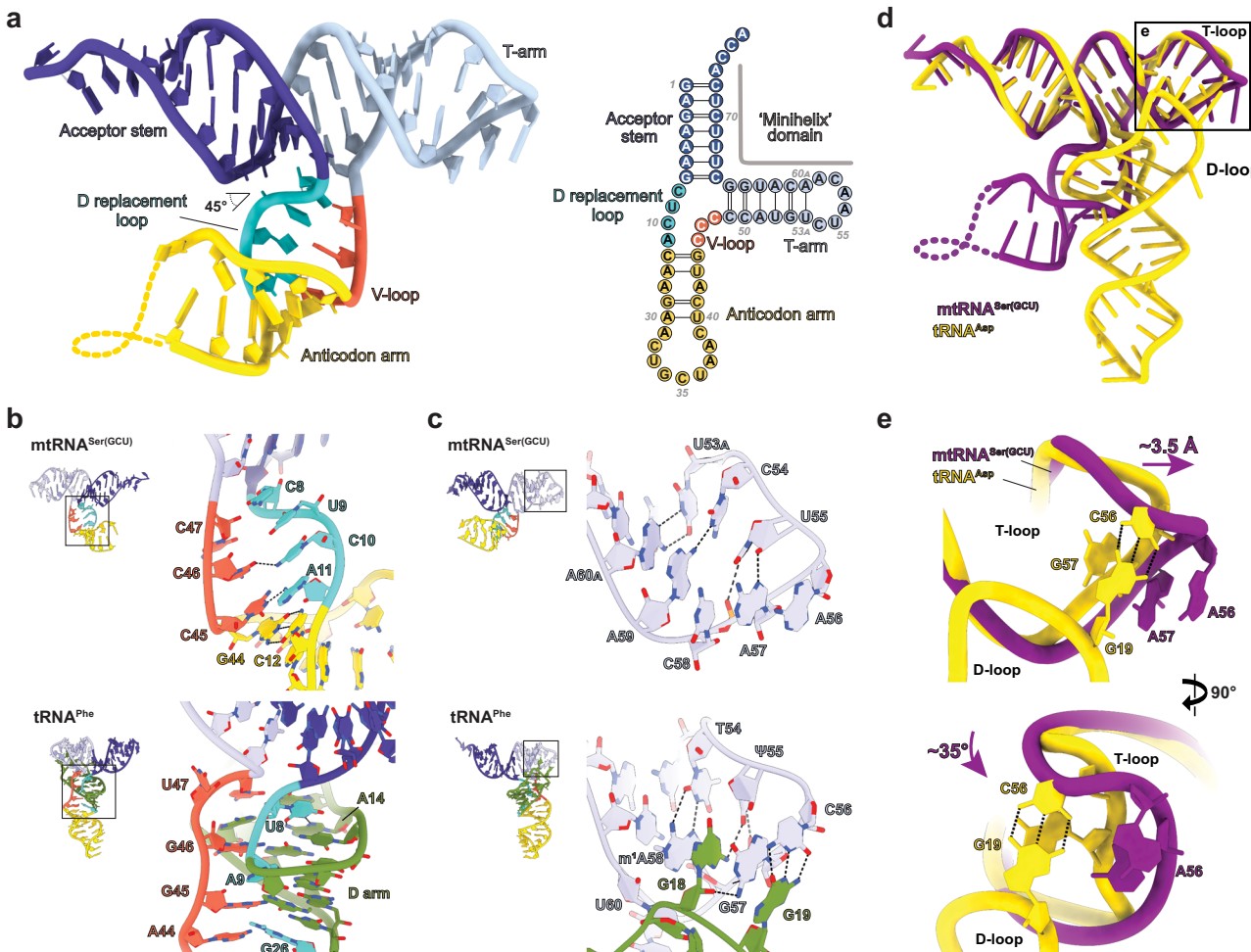

**Fig. 2 | The non-canonical structure of mtRNA^Ser(GCU).** **a** Tertiary (left) and secondary (right) structure of mtRNA^Ser(GCU). The portion of the anticodon arm missing in the reconstruction is indicated as yellow dashed line. **b** Details of the tRNA 'core' region in mtRNA^Ser(GCU) (top) and the canonical *S. cerevisiae* tRNA^Phe (bottom; PDB 4TRA) with view on the variable loop (orange). Color code as in **a**. The D-arm of tRNA^Phe, which is absent from mtRNA^Ser(GCU), is shown in green. **c** Details of the T-loop/elbow region in mtRNA^Ser(GCU) (top) and the canonical *S. cerevisiae* tRNA^Phe (bottom). Color code as in **a**. The D-loop of tRNA^Phe is shown in green. **d** Comparison of the structures of mtRNA^Ser(GCU) (purple) and the canonical *E. coli* tRNA^Asp (yellow; PDB 6UGG). The region enlarged in **e** is indicated. **e** Close-up view of the superimposed elbow regions from mtRNA^Ser(GCU) (purple) and *E. coli* tRNA^Asp (yellow), showing the shift and rotation of A56 in mtRNA^Ser(GCU) relative to the homologous C56 in canonical tRNAs as consequence of the additional T-stem pair U53_A:A60_A (see also **c**). Dashed lines indicate the H-bonds of the tertiary G19:C56 base pair which is found in all canonical tRNAs but absent from mtRNA^Ser(GCU).

type mtRNA^Ser(GCU) or a stabilized mtRNA^Ser(GCU)-TL construct ("TL" denotes a stabilized anti-codon loop; see "Methods" and Supplementary Fig. 4) to resolutions of 4.1 and 4.0 Å, respectively (Fig. 1d, e; Supplementary Figs. 5−7 and Supplementary Table 2). Both complexes show a well-defined density for a single tRNA molecule bound across the synthetase dimer interface. Due to the high intrinsic flexibility of the anticodon domain, the atomic model of the wild-type tRNA covers only ~65% of its sequence, comprising the complete acceptor-stem-T-arm ('minihelix') domain (Supplementary Fig. 7a). The stabilized mtRNA^Ser(GCU)-TL construct yielded an improved reconstruction that allowed modeling of an additional two-thirds of the anticodon domain (Fig. 1d, e). Docked onto both mSerRS subunits, the tRNA establishes three major interfaces: (1) with the active site entrance of subunit 1; (2) the C-tail and N-helix of subunits 1 and 2, respectively; and (3) the helical arm of subunit 2. Like other SerRS-tRNA^Ser complexes[24,25], the human mSerRS-mtRNA^Ser(GCU) interaction does not involve any contacts with the anticodon stem-loop. Notably, while both active sites are occupied by a SerSA molecule, the second tRNA binding site is incomplete in the cryo-EM reconstruction, with the N-helix and helical arm of subunit 1 not resolved in the EM densities (Fig. 1c). The absence of a second tRNA in the second binding

site of mSerRS is further supported by mass photometry data (Supplementary Fig. 7d), suggesting that in contrast to the bovine system[25], the human mSerRS dimer preferably binds a single mtRNA in solution. This asymmetry in the mSerRS-mtRNA^Ser complex indicates that its two tRNA binding sites are not equivalent once one tRNA is bound, an observation that has been made for canonical SerRSs as well[26–28].

## mtRNA^Ser(GCU) has lost the canonical tRNA fold

The canonical L-shape and tertiary structural core of tRNAs are nearly absolutely conserved throughout cellular life, as they provide functional determinants and a platform for interactions with other components of the translation machinery (Supplementary Fig. 1a−c)[5,8,9].

Our structures demonstrate the complete absence of a tRNA core in human mtRNA^Ser(GCU) (Fig. 2). The D-arm is eroded and replaced by a minimal four-nucleotide linker (the 'D replacement loop' ⁸CUCA¹¹) that forms loose pairs with a vestigial V-loop (⁴⁵CCC⁴⁷) and enters directly into an extended stack with the anticodon stem (Fig. 2a, b). Following the non-canonical A11·C45 pair, an additional C12:G44 Watson-Crick pair is inserted, extending the anticodon-stem to six instead of the canonical five base-pairs, consistent with previous predictions[29].

Strikingly, no tertiary interactions are formed between the 'D replacement loop' and acceptor-T-arm domain (Fig. 2a–c). As a consequence, the anticodon domain is not defined in the EM density of the wild-type tRNA and becomes only partially defined in the more stable mtRNA$^{Ser(GCU)-TL}$, although no stable tertiary interactions with the acceptor- or T-arm are observed (Fig. 2a). The anticodon- and acceptor arms hereby enclose an inter-stem angle of ~45° (Fig. 2a), which is notably smaller than the ~90–100° in canonical tRNAs (Supplementary Fig. 1b)[5]. Thus, mtRNA$^{Ser(GCU)}$ adopts an acute-angled 'Y' instead of the signature 'L'-shape of canonical tRNAs (Fig. 2d). However, the 'Y'-form appears to be intrinsically unstable and may be favored only in the complex with mSerRS, as suggested by the extreme conformational flexibility between anticodon- and acceptor arms in bovine mtRNA$^{Ser(GCU)}$ in solution[30]. Interestingly, it was recently shown that mtRNA$^{Ser(GCU)}$ is post-transcriptionally modified, with three consecutive 5-methylcytosines (m$^5$C) at positions C47, C49, and C50[31]. As suggested for cytoplasmic tRNAs, these modifications may stabilize mtRNA$^{Ser(GCU)}$ at the junction between V-loop and T-stem and thus account for the increased thermal stability of modified versus unmodified versions[32].

## Structural idiosyncrasies in the T-stem and T-loop

A radically reshaped 'T-arm' topology is another unexpected finding in mtRNA$^{Ser(GCU)}$ (Fig. 2c). The T-arm of canonical tRNAs is composed of a five-base-pair stem and the eponymous seven-nucleotide TψC-loop (T-loop)[5,7]. The latter adopts a compact U-turn structure, which is closed by a reverse Hoogsteen U(T)54-A58 pair and followed on its 3'-end by two bulged-out nucleotides (59 and 60) (Fig. 2c). Both the U-turn and bulge participate in tertiary interactions with the D-loop to define the tRNA's elbow structure and resulting L-shape (Supplementary Fig. 1a, b)[5,7]. Accordingly, the structure and dimensions of the T-arm are highly conserved across prokaryotic and eukaryote cytoplasmic tRNAs[5,9].

Free from all constraints imposed by tertiary interactions, the T-arm is substantially remodeled in mtRNA$^{Ser(GCU)}$ (Fig. 2c and Supplementary Fig. 8a). One of the most striking changes is the absence of bulged-out nucleotides in the 'T-loop'. A59 and A60$_A$ are both folded inwards, allowing a continuous base-stack from A56 through to A60$_A$ and into the T-stem without insertion of external bases. Moreover, the $^{54}$TψC$^{56}$ motif is replaced in mtRNA$^{Ser(GCU)}$ by $^{54}$CUA$^{56}$, on the one hand substituting the canonical C56 (in canonical tRNAs part of the invariant G19:C56 tertiary base-pair) for A56, and on the other causing the loss of the T-loop-defining U54-A58 reverse Hoogsteen pair. Notably, U55 is the only 'T-loop' nucleotide in mtRNA$^{Ser(GCU)}$ that is nearly invariant across vertebrate species (>98% conserved as part of a $^{55}$URR$^{57}$ motif; Supplementary Fig. 9), consistent with a crucial role in stabilizing the U-turn by a hydrogen-bond to the C58-phosphate (Fig. 2c). Overall, this creates a closed, self-contained loop topology that does not comply with the canonical definition of a T-loop but is instead structurally similar to the U-turns in tRNA anticodons and the IIa-loop of U2-snRNA (Supplementary Fig. 8b)[9,33]. (Despite this deviation, we continue to use the term 'T-loop' in mtRNA$^{Ser(GCU)}$ due to its evolutionary relationship with the T-loop of canonical tRNAs).

A further notable idiosyncrasy is the insertion of an additional stacking plane (U53$_A$:A60$_A$) into the T-stem of mtRNA$^{Ser(GCU)}$ (Fig. 2c). While structural and functional constraints limit the T-stem in canonical tRNAs to five base-pairs[5], it is extended to a unique six base-pairs in mtRNA$^{Ser(GCU)}$, elongating the T-arm by ~3.5 Å (Fig. 2d, e). Consequently, A56 at the outer tip of the U-turn is rotated by ~35° relative to the corresponding C56 in canonical tRNAs (Fig. 2e). Both changes are incompatible with the canonical tRNA structure and likely contingent on the prior or simultaneous erosion of the D-arm.

## Loss of canonical identity elements in mtRNA$^{Ser(GCU)}$

'Identity elements' in canonical aminoacylation systems are unique and highly conserved nucleotides embedded into the tRNA's structural scaffold, which are recognized by aaRSs to discriminate cognate from non-cognate tRNA[10,11]. Many of these elements are conserved from prokaryotes to the human cytoplasm and thus have remained virtually unchanged over billions of years of independent evolution. Among the most ancient of these is the long V-arm of canonical tRNA$^{Ser}$, which is recognized by bacterial, archaeal, and eukaryote cytoplasmic SerRSs alike[24,26]. In view of the strong selective pressure to maintain identity elements, it is striking that animal mtRNA$^{Ser(GCU)}$ have completely lost the long V-arm[18,29,34], only maintaining a vestigial V-loop ($^{45}$CCC$^{47}$ in humans) (Fig. 2a and Supplementary Fig. 1c, d). Sequence analysis further shows that, apart from the anticodon loop, no position in mtRNA$^{Ser(GCU)}$ is absolutely conserved across vertebrates, and even among mammals only five positions, A2:U71, C49:G65, and U55, are (nearly) invariant (Supplementary Fig. 9a, b). This raises the question of how animal mitochondrial serylation systems compensated the loss of conserved sequence identity elements and ensured specificity between mtRNA$^{Ser(GCU)}$ and mSerRS[21,25].

Our structures show how the loss of identity elements in mtRNA$^{Ser(GCU)}$ was accompanied by an extensive metamorphosis of the SerRS-tRNA interaction (Fig. 3). Most notably, mSerRS binds mtRNA$^{Ser(GCU)}$ exclusively at its minihelix domain, which comprises the coaxially stacked acceptor-stem and T-arm (Fig. 3a and Supplementary Fig. 10a, b). In stark contrast to previously described aaRS-tRNA interactions, neither the V-loop nor the flexible anticodon domain contribute to binding (Fig. 3b and c). Whereas the N-terminal helical arm of bacterial SerRSs forms a critical recognition site for the long V-arm of canonical tRNA$^{Ser}$[24,26], this interface in human mSerRS does not exhibit the positive electrostatic surface potential necessary to establish these interactions (Fig. 3d, e). Instead, the helical arm of mSerRS exclusively contacts the tRNA's idiosyncratic 'T-loop', accommodating the U-turn in a deep, positively charged binding pocket framed by Lys110, Arg118, Arg139, Arg143, and Arg146 (Fig. 3d). A new interface is formed by the mSerRS-specific N-helix and C-tail with the tRNA's T-stem, filling a gap found in canonical SerRS-tRNA complexes and creating a continuous binding interface along the entire length of the acceptor-T-arm domain (Fig. 3a and Supplementary Fig. 10a, b).

Of the 22 mSerRS residues that interact with the tRNA (Supplementary Fig. 10a), most contacts are with the phosphate-sugar backbone. Only five base-specific interactions may be formed at the entrance to the active site with the CCA-end, 'discriminator base' (A73), and the first acceptor-stem pair (Fig. 4). Notably, binding of the tRNA induces a major conformational rearrangement in the motif 2 loop (Supplementary Fig. 10c), which involves a ~180° flip of Glu332 relative to its conformation in free mSerRS-SerSA and allows the insertion of residues 334–340 into the major groove of the tRNA acceptor stem (Fig. 4b). This conformational transition is similar to that observed in the *T. thermophilus* SerRS-tRNA$^{Ser}$ complex[35], however, the resulting interactions differ substantially between the two complexes. Most notably, the flipped Glu332 of mSerRS does not contact the discriminator base A73 (G73 in the bacterial tRNA$^{Ser}$), and residue Phe262 of *T. thermophilus* SerRS, which provides selectivity for pyrimidine bases in positions 68 and 69 of tRNA$^{Ser}$, is not present in mSerRS (Supplementary Fig. 3). The reach of the motif 2 loop is thus reduced, preventing formation of any of those interactions that confer sequence-preferences in positions 2:71 and 3:70 by the bacterial SerRS[26,35]. Consequently, the A2:U71 pair has no direct contact with the synthetase (Fig. 4b), despite being the only acceptor-stem pair that is conserved in mtRNA$^{Ser(GCU)}$ (Supplementary Fig. 9), and mutation to G2:C71 has no adverse effect on aminoacylation by human or bovine mSerRS (Fig. 5a, b)[34]. Instead, new hydrogen bonds are formed by the main-chain amide groups of Thr335 and Gly336 in the shortened motif 2 loop with N7 and possibly O6 of G1 (~3 Å). A further contact may occur between Asn334 and A73; however, neither Asn334, nor the G1:C72 base-pair or A73 are conserved in mammalian mtRNA$^{Ser(GCU)}$

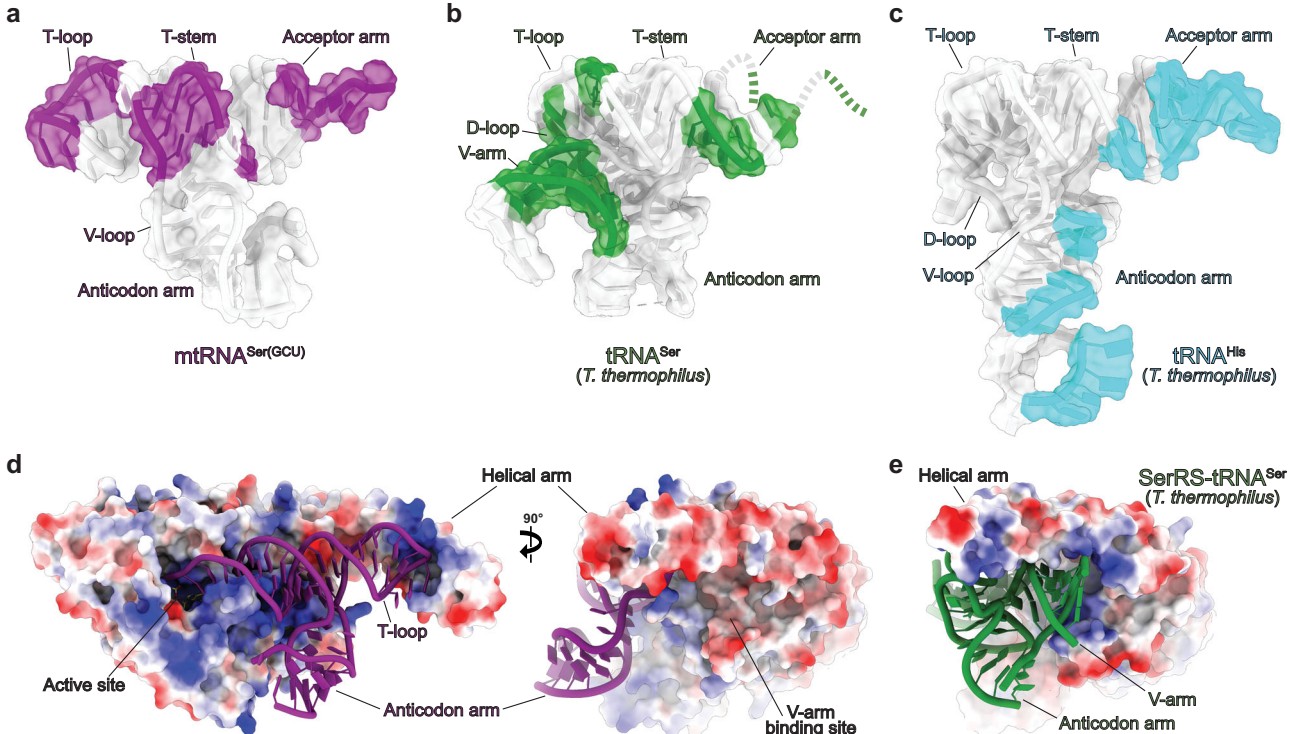

**Fig. 3 | The reshaped interface between mSerRS and mtRNA^Ser(GCU).** **a** Surface regions (purple) of mtRNA^Ser(GCU) in contact with mSerRS. Regions outside of the interface are shown in white. **b** Surface regions (green) of *T. thermophilus* tRNA^Ser forming polar contacts with bacterial SerRS (PDB 1SER)[26, 35]. **c** Surface regions (cyan) of *T. thermophilus* tRNA^His in contact with HisRS (PDB 4RDX)[74], showing a distribution across anticodon-D-arm and acceptor-T-arm domains typical for anticodon-recognizing aaRS-tRNA complexes. **d** Electrostatic surface potential of the mSerRS-mtRNA^Ser(GCU) complex (shown as a three-color gradient scheme from −5 to +5 kcal mol$^{-1}$ e$^{-1}$ (red, negative; white, neutral; blue, positive)). The tRNA (purple) is shown in cartoon presentation. The V-arm binding site of canonical SerRSs that is unoccupied and degenerated in mSerRS is indicated on the right. **e** Electrostatic surface potential of the *T. thermophilus* SerRS-tRNA^Ser complex (PDB 1SER)[26] (viewed as in the right panel in **d**).

(Supplementary Figs. 9 and 10d), and the A73G mutation has no negative effect on aminoacylation (Fig. 5a, b). Consistent with bio-chemical data for bovine mtRNA^Ser(GCU)[25,34], this suggests that the acceptor arm of human mtRNA^Ser(GCU) does not contain major sequence identity elements for its recognition by mSerRS.

Like the acceptor stem, most of the T-stem is poorly conserved in mtRNA^Ser(GCU) (Supplementary Fig. 9). While residues from the N-helix and C-tail form polar interactions with the sugar-phosphate backbone, no base-specific interactions are made with the T-stem minor groove (Fig. 4c). This is supported by our mutational analysis, which shows that substitutions of the C50:G64 and A51:U63 base-pairs have no adverse effect on charging (Fig. 5a, b). On the other hand, deletions of the N-helix or the C-tail nearly abolish the aminoacylation activity of bovine mSerRS[21]. Taken together, this suggests that the N-helix and the C-tail provide an important, but non-specific binding site for mtRNA^Ser(GCU), possibly compensating for the loss of the V-arm interface without contributing to the specificity of tRNA binding through direct, sequence-specific readout. The m$^5$C modifications in C49 and C50[31], which lie on the major groove side of the T-stem and thus opposite to the binding interface, may hereby have a positive, but likely indirect impact by stabilizing the local T-stem architecture.

### mSerRS recognizes mtRNA^Ser(GCU) by its unique T-arm topology

The mSerRS helical arm interface is notably distinct from the other two interfaces, as it involves the idiosyncratic 'T-loop' (Fig. 4a, d), a struc-tural feature of mtRNA^Ser(GCU) found in no other tRNA. Relative to free mSerRS, substrate binding induces a major conformational rearran-gement in the helical arm (Supplementary Fig. 10e). The unpaired and exposed A56 is inserted deep into a helical arm binding pocket, next to Val117 (Fig. 4d). The importance of this interface is supported by

aminoacylation kinetics: The A56C transversion mutation, which reintroduces a canonical C56, nearly abolished charging (Fig. 5a). By contrast, the A56G transition mutation had no effect, suggesting a preference for purine bases, but without discriminating between A or G. Surprisingly, individual Ala-mutations of R118A, R139A, and R143A in the A56 binding pocket affected tRNA charging only weakly. The triple mutation ('3xRA'), however, significantly reduced charging, demon-strating their collective importance for tRNA binding (Fig. 5c).

A mutational study of the bovine mSerRS-mtRNA^Ser(GCU) inter-action suggested an important role for Arg146 in tRNA recognition[21]. The structure of the human complex shows that Arg146, which is highly conserved across vertebrate mSerRSs (Supplementary Fig. 11), is not part of the A56 binding pocket but protrudes into the concave backside of the 'T-loop' to form a net-work of long-range (4–6 Å) electrostatic interactions with the phosphate-oxygens from three contiguous nucleotides (C54-A56) (Fig. 4d). The single mutation of Arg146 to alanine is sufficient to nearly abolish aminoacylation by human mSerRS (Fig. 5c), sug-gesting an important contribution to human mtRNA^Ser(GCU) binding and recognition. A comparable reduction in charging resulted from the U55C mutation in mtRNA^Ser(GCU) (Fig. 5a), which disrupts the highly conserved tertiary hydrogen-bond to the C58-phosphate (Fig. 2c). Since the U55 base is not in direct contact with the syn-thetase, its contribution is most likely indirect by stabilizing a 'T-loop' backbone topology that is recognizable for Arg146 and allows the insertion of the U-turn into the helical arm binding pocket (Fig. 4d).

To further understand the importance of the T-arm structural context for mtRNA^Ser(GCU) recognition, we tested two tRNA variants in which either the T-stem was reduced to the canonical five base-pairs

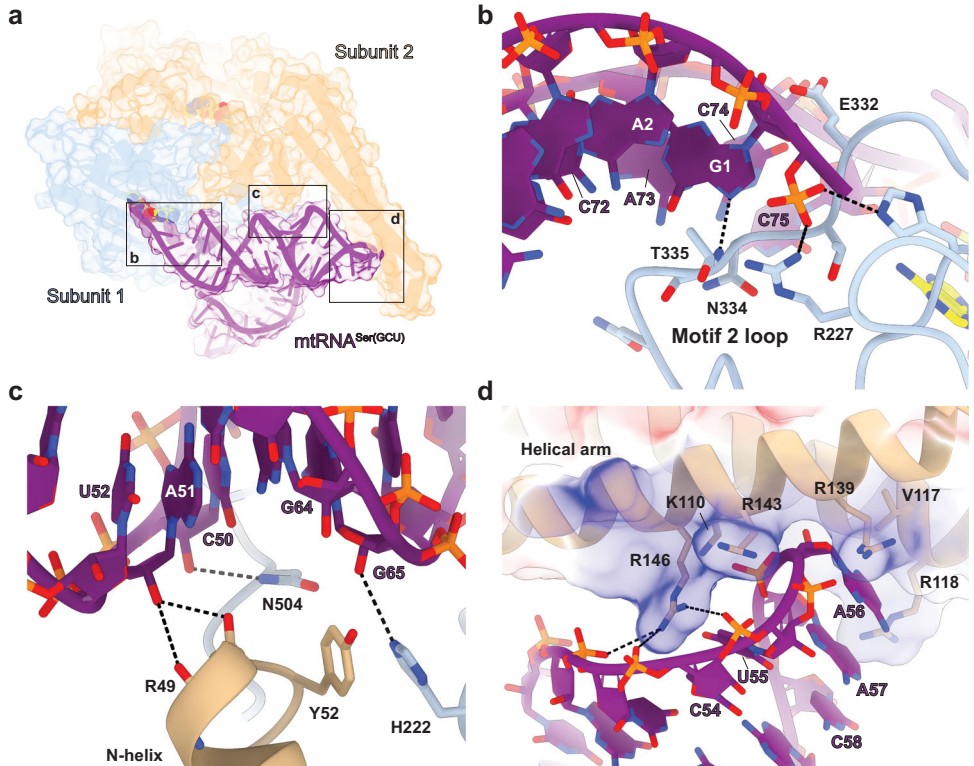

**Fig. 4 | Binding interfaces between mtRNA^Ser(GCU) and mSerRS. a** Overview of the mSerRS-mtRNA^Ser(GCU) complex with regions enlarged in **b**, **c**, and **d** indicated. **b** Close-up view of the interactions between the acceptor arm of mtRNA^Ser(GCU) (purple) and the motif 2 loop (blue) at the entrance to the active site. **c** Close-up view of the interactions of the T-stem (purple) with the N-helix (wheat) and C-tail (blue) of mSerRS. C50 is modified to m^5C in human mtRNA^Ser(GCU)[31], with the 5-methyl group pointing away from the synthetase interface. **d** Close-up view of the interface between the 'T-loop' (purple) and the helical arm (wheat). Residue side chains of mSerRS are shown as sticks. Arg118, Arg139, and Arg143 are not well resolved in the EM densities and shown as their most likely rotamers. The electrostatic surface potential of mSerRS is shown transparently as a three-color gradient scheme from −5 to +5 kcal mol⁻¹ e⁻¹ (red, negative; white, neutral; blue, positive).

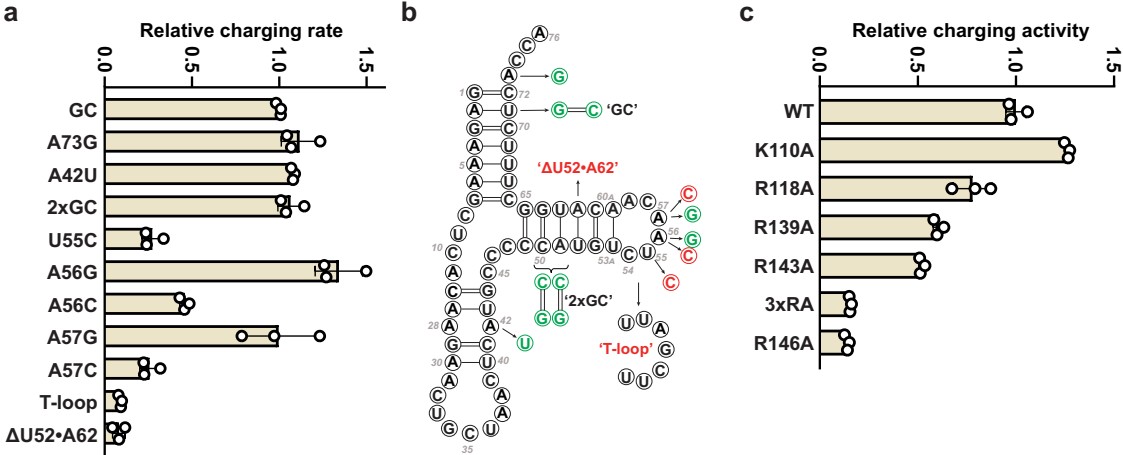

**Fig. 5 | Mutational and kinetic analysis of the mSerRS-mtRNA^Ser(GCU) interaction. a** Summary of charging experiments for mtRNA^Ser(GCU) mutants by wild-type mSerRS. **b** Summary of mtRNA^Ser(GCU) mutants used in **a**. 'GC' denotes stabilized 'wild-type' tRNA (see Supplementary Fig. 4 and "Methods"). **c** Charging kinetics for mSerRS variants on mtRNA^Ser(GCU). 3xRA denotes the R118A, R139A, R143A triple mutant. Data are presented as mean values +/− SD from triplicate experiments. Individual data points are shown as open circles. See also Supplementary Tables 3 and 4.

(ΔU52:A62), thereby reversing the mtRNA^Ser(GCU)-specific extension and rotation of the 'T-loop', or the unique 'T-loop' itself was replaced by that from a canonical tRNA (Fig. 5b). Both mutations abolished charging (Fig. 5a). These results further support the idea that it is the non-canonical topology of the 'T-arm' that serves as the critical identity determinant in the mSerRS-mtRNA^Ser(GCU) interaction.

## Discussion

Here we present structures of mitochondrial SerRS captured in complex with its cognate substrate, mtRNA^Ser(GCU), providing molecular insight into the first step of human mitochondrial protein synthesis. These structures, together with mutational and kinetic data, reveal how the most degenerated human mtRNA is specifically recognized

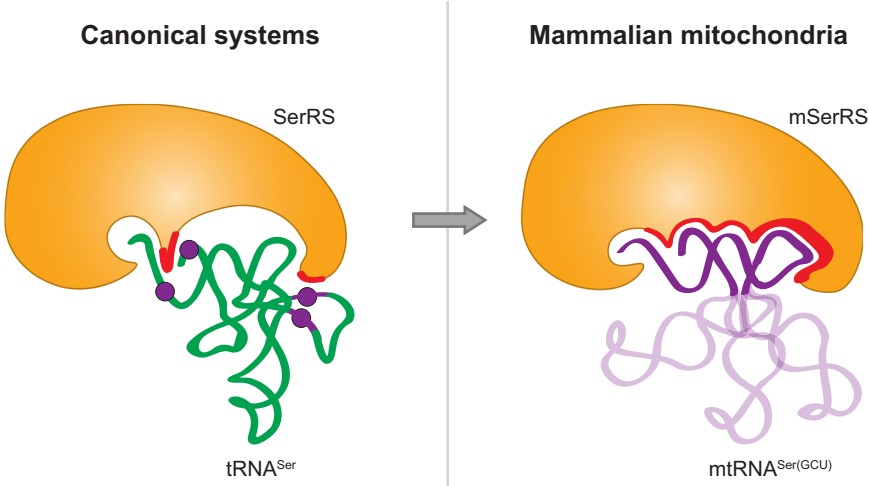

**Canonical systems**

SerRS

tRNA^Ser

**Mammalian mitochondria**

mSerRS

mtRNA^Ser(GCU)

**Fig. 6 | Divergence and rewiring of tRNA^Ser identity rules in animal mitochondrial gene expression.** In the ancestral canonical serylation systems (left), SerRS relies on discrete sequence identity elements (purple) inserted into the universal structural scaffold of the canonical tRNA. In the derived mammalian mitochondrial system (right), the divergent structural scaffold itself becomes the main identity-defining feature that is recognized by mSerRS by shape- and charge-complementarity through a remodeled and enlarged binding interface.

and aminoacylated by its cognate synthetase, and provide a mechanistic basis to understand the principles of co-evolution between nuclear-encoded proteins and their rapidly evolving mitochondrially encoded tRNA substrates.

Since their discovery more than 40 years ago, the degenerated structures and functional specialization of mitochondrial tRNAs have remained poorly understood[3,18,36]. The D-armless mtRNA^Ser(GCU) in particular attracted attention as the shortest and most divergent mammalian mtRNA[12,18,29,34,37]. Previous attempts to model mtRNA^Ser(GCU) were invariably based on the premise that the erosion of the D-arm had to be compensated by novel tertiary interactions between the 'D replacement loop' and the T-loop, thereby stabilizing a boomerang-like overall fold[29,37]. Our structural analysis reveals that no such compensatory interactions exist in human mtRNA^Ser(GCU) (Fig. 2). Neither the 'D replacement loop', which is part of an extended anticodon stem, nor the remodeled, non-canonical 'T-loop' provide bulged-out nucleotides that could participate in tertiary interactions, making mtRNA^Ser(GCU) the only tRNA known to completely lack a tertiary structural core. Although the observed Y-shape of mtRNA^Ser(GCU) may be favored only in the context of the mSerRS complex, earlier structure probing data suggest that the 'T-loop' in its free state remains without stable tertiary interactions[30,37,38]. The result is an exceptionally high internal flexibility in mtRNA^Ser(GCU) in solution[30], which allows the anticodon domain to rotate relative to the acceptor-T-arm domain with inter-stem angles ranging from ~45° to ~120°.

In notable contrast to the degeneration of D-arm and tertiary core stands the extensive reconstruction of the T-arm, which was expanded in mtRNA^Ser(GCU) beyond the structural constraints imposed on canonical tRNAs (Fig. 2). While unique among the 22 human mtRNAs, its features are conserved in mtRNA^Ser(GCU) throughout the vertebrate lineage (Supplementary Fig. 9d), suggesting that they are maintained as a result of positive selection pressure. Importantly, this implies that the divergence of mammalian mtRNAs is not driven exclusively by the degenerative forces of mutation accumulation and genetic drift[39,40], but also by adaptive evolution leading to functionally relevant structural innovations.

Together with data from the bovine system[21,25], our analyses reveal that the divergence of animal mtRNA^Ser(GCU) was accompanied by a radical rewiring of its interactions with mSerRS (Fig. 6). Unlike any other known aaRS-tRNA complex, canonical or mitochondrial[10,41,42], mSerRS contacts mtRNA^Ser(GCU) exclusively at the acceptor-T-arm

(minihelix) domain, making the interaction essentially independent from the extreme conformational flexibility in the tRNA's core region (Fig. 3a–c). The peptide extensions on both termini of mSerRS thereby expand the ancestral T-arm interface, establishing a continuous interface along the entire length of the minihelix domain. Notably, sequence-specific interactions, like those found in canonical systems[24], play only a secondary role in the form of purine-pyrimidine bias, as reflected in the high sequence variability in mtRNA^Ser(GCU) throughout its mSerRS binding interface (Supplementary Fig. 9). Instead, specificity is established almost exclusively through shape- and charge-complementarity, with the unique 'T-arm' topology of mtRNA^Ser(GCU) serving as the major identity element, 'read out' by strategically positioned arginine residues in the conformationally adaptable helical arm.

The identity set of the human mitochondrial recognition system shows virtually no overlap with its ancestral bacterial SerRS-tRNA^Ser counterpart[24,26,35], with the major recognition sites shifted from the long V-arm, D-arm, and acceptor stem in bacterial tRNA^Ser to the 'T-loop' in mtRNA^Ser(GCU) (Fig. 6). Analogous shifts occurred throughout animal mitochondrial aminoacylation systems[43–45], most notably between human mtRNA^Ala and mAlaRS[14,46], which have likewise led to the loss of ancient sequence identity elements. Previously interpreted as merely the result of simplification and relaxed functional constraints on mitochondrial translation[3,39,47], our results suggest that they may reflect a fundamental shift of molecular recognition rules in mitochondrial aaRS-tRNA interactions (Fig. 6), in which structural heterogeneity among mtRNAs becomes an important determinant for molecular identity based on backbone topology, conformational state, and intrinsic flexibility.

The properties of the mitochondrial serylation system provide new insight into the evolutionary mechanisms and long-term functional stability of mitochondrial gene expression. Theory predicts that the mitochondria-encoded tRNA genes of bilaterian animals undergo continuous mutational erosion by 'Muller's ratchet', leading to an irreversible decline in mitochondrial fitness over time[39,40,47,48]. Paradoxically, despite high mutation rates, no such functional decline is observed[40], implying that mechanisms exist to mitigate the functional 'meltdown' of mtRNAs and their interactions. We propose that constructive coevolution between mtRNAs and nuclear-encoded mt-aaRSs plays an important, previously unrecognized role, involving adaptive changes in both interaction partners. In particular, the identity-defining structure elements in mtRNA^Ser(GCU) are not merely vestiges

of a mutationally eroded tRNA core (as would be expected from the 'nuclear compensation hypothesis'), but represent mitochondria-specific innovations, unique to mtRNA$^{Ser(GCU)}$ and essential for its shape-selective readout by mSerRS. Once introduced, these new identity determinants became fixed in the vertebrate lineage (Supplementary Figs. 9 and 11), thereby reestablishing long-term functional stability despite continuous mutational insults. This suggests that mutational robustness may be an important driving force in the adaptive rewiring of mitochondrial aaRS-tRNA interactions, offering a mechanistic explanation for the ability of mtRNAs to evolve rapidly in highly epistatic fitness landscapes[14,49], without long-term functional decline or the need for continuous compensatory coevolution in nuclear-encoded mitochondrial synthetases[50,51].

While the focus of this study lies on mtRNA$^{Ser(GCU)}$, its structure and recognition, it is important to note that the human mtDNA encodes a second serine isoacceptor, mtRNA$^{Ser(UGA)}$. Notably, mtRNA$^{Ser(UGA)}$ shares virtually no sequence or structural homology with mtRNA$^{Ser(GCU)}$, raising the question of how both mtRNAs are recognized by a single synthetase[52]. According to studies on the bovine system, mSerRS evolved distinct mechanisms to recognize each of its divergent mtRNA$^{Ser}$ substrates[21,25]. Our results support such dual recognition also for the human mSerRS-mtRNA$^{Ser}$ interactions and suggest that distinct structural features serve as idiosyncratic identity elements in mtRNA$^{Ser(GCU)}$ and mtRNA$^{Ser(UGA)}$. How such dual recognition is achieved at the molecular level is currently unknown and will require high-resolution structural insight into the mSerRS-mtRNA$^{Ser(UGA)}$ complex.

Taken together, our results present an important step toward understanding in detail the structural and mechanistic innovations underlying mammalian mitochondrial protein synthesis. They provide a molecular basis to understand the functional specialization of mitochondrial tRNA recognition and present a vantage point for future studies into the principles of mito-nuclear coevolution and regulation in animal mitochondrial gene expression.

## Methods

### Cloning, expression, and purification of human mSerRS constructs

The gene coding for human mSerRS lacking the mitochondrial targeting sequence (aa 1–30) was cloned into BamHI/XhoI sites of pGEX-6P1 vector, resulting in the expression with an N-terminal GST-Tag (primers used for PCR amplification and cloning are summarized in Supplementary Table 5). *E. coli* BL21(DE3) cells containing the mSerRS plasmid were grown in LB medium at 37 °C to an OD of 0.7. Cultures were then allowed to cool down to room temperature (23 °C) and expression was induced by addition of IPTG at a final concentration of 0.5 mM. Cells were harvested after shaking 16 h at RT. GST-tagged mSerRS was purified by GSH-sepharose beads (Qiagen), followed by the cleavage of the GST-tag by PreScission protease cleavage, HiTrap Heparin (GE Healthcare), and a HiLoad 16/60 Superdex 200 column (GE Healthcare) equilibrated in 20 mM HEPES pH 7.5, 150 mM NaCl, and 1 mM DTT. Purified mSerRS was concentrated and stored at −80 °C. Mutant proteins were constructed by site-directed mutagenesis and purified using the same protocol. The quality of protein preparations was validated by SDS-PAGE analysis.

### Synthesis of 5'-O-[N-(L-seryl)sulfamoyl] adenosine (SerSA)

The seryl-adenylate analog 5'-O-[N-(L-seryl)sulfamoyl] adenosine (SerSA) was synthesized essentially as described previously[53], with slight variations. Briefly, 2′,3′-O-isopropylideneadenosine was heated to reflux for 5 h with a molar excess of bis(tributyltin) oxide giving its 5′-O-tributyltin ether[54]. The mixture was then treated with an excess of sulfamoyl chloride. After column chromatography, the intermediate compound was reacted with the *N*-Hydroxysuccinimide (NHS) ester of methyl (S)-(−)-3-Boc-2,2-dimethyl-4-oxazolidinecarboxylate in the presence of 1,8-Diazabicyclo(5.4.0)undec-7-ene (DBU), followed by

global acidic deprotection in aqueous TFA, yielding SerSA (Supplementary Fig. 12). The final product was purified by HPLC.

### In vitro transcription of tRNAs

Genes encoding mtRNA$^{Ser(GCU)}$ variants were either cloned into pUC-19 vector or purchased as synthetic oligos with the tRNA-coding region under the control of a T7 RNA polymerase promoter. Mutant genes were generated by site-directed mutagenesis following the Quik-Change protocol (Stratagene). All DNA-templates for in vitro transcription were amplified by PCR using forward and reverse primers complimentary to the T7 promoter and the 3' end of the tRNA gene, respectively. Run-off transcription reactions were performed in 40 mM Tris-HCl pH 8.0, 25 mM NaCl, 25 mM MgCl$_2$, 2 µg/mL yeast pyrophosphatase (Roche), 1 mM Spermidine, 5 mM DTT, 18 mM GMP, 4 mM each of ATP, CTP, GTP, and UTP with 75 µg/mL T7 polymerase and DNA template at 37 °C for 6 h. Reactions were stopped by phenol/chloroform extraction followed by purification of the tRNA by 12% denaturing PAGE. tRNA was eluted from the crushed gel slices in buffer containing 200 mM NaOAc, 20 mM Tris/HCl, 5 mM EDTA (pH 5.3). The eluted tRNA was then annealed by first heating to 80 °C, followed by gradual cooling to 20 °C at a rate of 2°/min. At 60 °C MgCl$_2$ was added to a final concentration of 7.5 mM. The tRNA was finally ethanol-precipitated, taken up in RNase-free water and stored at −80 °C.

The mtRNA$^{Ser(GCU)}$ is inherently unstable and prone to misfolding. As it was shown previously that the acceptor stem does not contribute to tRNA selectivity in the bovine mitochondrial serylation system[25,34], we altered the human mtRNA$^{Ser(GCU)}$ construct used in kinetic studies to contain three consecutive G:C in first three acceptor stem base-pairs to improve the efficiency of in vitro transcription and to increase the stability of the correctly folded tRNA. Moreover, SerRSs do not recognize the anticodon of their tRNA$^{Ser}$ substrates[34]. To improve the stability of tRNA constructs used for structural analysis, we introduced a more stable GAAA tetraloop in lieu of the tRNA anticodon. Both alterations increased the overall yield of chargeable mtRNA$^{Ser(GCU)}$ without reducing charging rates, suggesting that they promoted correct folding (Supplementary Fig. 4).

### Active site titration assay

The concentration of active sites was determined at RT (25 °C) in 40-µl reactions containing two different concentrations (5 and 10 µM) of human mSerRS, 20 mM L-serine, 22 nM [γ-$^{32}$P]-ATP, in assay buffer (100 mM HEPES pH 7.5, 20 mM KCl, 10 mM MgCl$_2$, 2 mM DTT, and 2 mg/mL yeast pyrophosphatase (Roche)). Reactions were initiated by adding enzyme to the assay solution in 96-well low-profile PCR plates. At different time points 5 µl reaction mix were quenched into PVDF MultiScreen filter plates (0.45 µm pore size hydrophobic, low-protein-binding membrane; Merck Millipore) containing 20 µl of 7% HClO$_4$ and 80 µl of 10% charcoal slurry. Following the last time point, the slurry was mixed by pipetting and centrifuged into a 96-well flexible PET microplate (PerkinElmer) containing 150 µL of Supermix scintillation mixture (PerkinElmer). The plate was counted on a 1450 MicroBeta Microplate Scintillation and Luminescence Counter (PerkinElmer). Kinetic data were analyzed using GraphPad Prism 8 (GraphPad Software, Inc.).

### In vitro aminoacylation

Aminoacylation reactions were carried out in an assay solution containing 50 mM HEPES pH 7.5, 60 mM KCl, 10 mM MgCl$_2$, 4 mM ATP, 5 mM DTT, 4 µg/mL yeast pyrophosphatase (Roche), 1 mM Spermine, 10 µM cold L-serine, and 5 µM [$^3$H]-serine (1 mCi/mL). Varying amounts of tRNA were initially mixed with assay solution, and the reaction was initiated by addition of human mSerRS (0.5 µM). At varying time intervals, 5-µL aliquots were removed and applied to a MultiScreen 96-well filter plate (0.45 µm pore size hydrophobic, low-protein-binding membrane; Merck Millipore), pre-wetted with quench solution

(0.5 mg/mL salmon sperm DNA, 0.1 M EDTA, 0.3 M NaOAc (pH 3.0)). After all time points were collected, 100 μL of 20% (w/v) trichloroacetic acid (TCA) was added to precipitate nucleic acids. The plate was then washed four times with 200 μL of 5% TCA containing 100 mM cold serine, followed once by 200 μL of 95% ethanol. The plate was then dried, followed by deacylation of bound tRNAs by addition of 70 μL of 100 mM NaOH. After 10 min incubation at RT, the NaOH-solution was centrifuged into a 96-well flexible PET microplate (PerkinElmer) with 150 μL of Supermix scintillation mixture (PerkinElmer). After mixing, the radioactivity in each well of the plate was measured in a 1450 MicroBeta Micoplate Scintillation and Luminescence Counter (PerkinElmer).

## Mass photometry

Mass photometry[55] experiments were performed in binding buffer composed of 20 mM HEPES pH 7.0, 100 mM NaCl, 7.5 mM MgCl$_2$, and 1 mM DTT. Data were acquired at final concentrations of 25 nM mSerRS, 125 nM mtRNA$^{Ser(GCU)}$, and 250 nM SerSA using a Refeyn One™ mass photometry system (Refeyn Ltd, Oxford, UK). The resulting video data were analyzed using DiscoverMP software (Refeyn Ltd, Oxford, UK). Raw contrast values were converted to molecular mass using a standard mass calibration, and binding events combined in 2.5 kDa bin width. Binding events below 40 kDa were indistinguishable from background. Detection settings were adjusted according to the specific visualization requirements and with a background reading of buffer alone.

## tRNA sequence analysis

Sequences for genes encoding mitochondrial, prokaryotic or eukaryote cytoplasmic tRNAs were retrieved from tRNAdb/mitotRNAdb (http://trna.bioinf.uni-leipzig.de/) and the genomic tRNA database (GtRNAdb; http://gtrnadb.ucsc.edu/). Sequence alignments of tRNA genes were performed using the ClustalW function of the Molecular Evolutionary Genetics Analysis (MEGA 7.0) software[56]. Misaligned regions were curated manually based on structural characteristics of the tRNAs (i.e., corrected for alignments of stem- and loop regions).

## Crystallization and X-ray data collection

Prior to crystallization, purified human mSerRS at a concentration of 10–12 mg/mL was mixed with 2 mM of SerSA. Initial high-throughput crystallization screens were performed by the sitting-drop vapor diffusion method using a Mosquito liquid transfer robot (TTP Labtech). Each drop contained 200 nL protein and 200 nL of reservoir solution and was equilibrated against 40 μL of reservoir solution. Single crystals grew after 2 days at 23 °C in 0.1 M Bicine (pH 9.0), 10 mM Spermidine, 8% PEG 6000. Crystals were cryo-protected with 10–15% (v/v) glycerol added to the reservoir solution and flash-frozen in liquid nitrogen. X-ray diffraction data were collected on beamline 12-2 at the Stanford Synchrotron Radiation Lightsource (SSRL) at 100 K and a wavelength of 0.9740 Å. Diffraction images were processed with the XDS package[57]. X-ray diffraction data statistics are summarized in Supplementary Table 1.

## X-ray structure determination and refinement

The mSerRS-SerSA complex crystallized in space group R3 with 2 molecules per asymmetric unit (AU), corresponding to the biological homo-dimeric assembly. The phase problem was solved by the Molecular Replacement (MR) method using PHASER[58] and the crystal structure of bovine mSerRS (PDB 1WLE)[21] as search model. Model building and refinement were performed in iterative cycles using Coot and Phenix-1.20.1[59,60]. The final model of the human mSerRS-SerSA complex was refined to 2.9 Å resolution with R$_{work}$ = 27.13% and R$_{free}$ = 30.94%. Each active site in the mSerRS homodimer is occupied by the SerSA ligand. A summary of refinement statistics is given in Supplementary Table 1.

## Complex reconstitution for cryo-EM analysis

Human mSerRS and tRNAs were purified individually as described above. Prior to the addition of tRNA, mSerRS was incubated with a 10-fold molar excess of SerSA ligand for 15 min at room temperature. The mSerRS-SerSA complex was then mixed with either mtRNA$^{Ser(GCU)}$ or mtRNA$^{Ser(GCU)-TL}$ at a protein:tRNA molar ratio of 1:1.5 (corresponding to a 3-fold molar excess of tRNA over mSerRS dimer) and incubated at room temperature for 20 min. The complex was loaded onto a Superdex 200 Increase 10/300 GL column (GE Healthcare) equilibrated in 20 mM HEPES pH 7.0, 100 mM NaCl, 7.5 mM MgCl$_2$, and 1 mM DTT. For both tRNA variants, the mSerRS-SerSA-tRNA complex eluted in a single peak. Peak fractions containing the highest concentration of the complexes were collected and immediately used for cryo-EM sample preparation.

## Cryo-EM sample preparation

Human mSerRS in complex with mtRNA$^{Ser(GCU)}$ or mtRNA$^{Ser(GCU)-TL}$ was diluted to a concentration of 0.75 mg/ml and samples containing tRNA mutants were mixed with 0.05% v/v Lauryl Maltose Neopentyl Glycol (Anatrace) immediately prior to plunge freezing. UltrAuFoil R1.2/1.3 300-mesh grids (Quantifoil) were plasma cleaned in a Solarus plasma cleaner (Gatan, Inc.) with a 75% nitrogen, 25% oxygen atmosphere at 15 W for 7 s. Cryo-EM grids were prepared by application of 4 μL protein sample at 4 °C in 95% humidity. The grids were manually blotted for 4–5 s using Whatman No. 1 filter paper, followed by plunge freezing in liquid ethane.

## Cryo-EM data acquisition

Cryo-EM data were collected on a Talos Arctica TEM (Thermo Fisher) operating at 200 keV in counting mode equipped with a K2 Summit direct electron detector (Gatan, Inc.). Data collection was automated using the Leginon data collection software[61]. Movies were collected at a nominal magnification of ×36,000 with a physical pixel size of 1.15 Å pixel⁻¹. The total number of movies for the tRNA$^{Ser(GCU)-TL}$ and tRNA$^{Ser(GCU)}$ datasets were 2498 and 1330, respectively, consisted of 200 ms frames and a total exposure time of ~11.8 s, resulting in a cumulative exposure of 66 electrons/Å$^2$. Movies were acquired using a nominal defocus range of 0.8–1.2 μm. Micrographs for the mSerRS-tRNA$^{Ser(GCU)-TL}$ complex were collected at 0, 20, and 40 degrees alpha tilt. Preprocessing was performed in real time using Warp[62] in order to monitor data quality. Particle stacks from Warp were input to CryoSPARC[63] for 2D classification (50 classes, 65% inner radius window) to assess variety of collected views.

## Human mSerRS-mtRNA$^{Ser(GCU)}$ cryo-EM data processing

Movies for mSerRS-tRNA$^{Ser(GCU)}$ were collected during two sessions, referred to as dataset 1 and dataset 2, which were merged at final stages of the refinement. Movies were aligned and dose-weighted using MotionCor2[64] in RELION 3.1[65,66] on 5 × 5 patches with an applied B-factor of 150. CTF estimation was performed using Gctf[67] on the non-dose weighted aligned images. The Laplacian of Gaussian picker[68] was used to pick particles and the resulting picks were extracted binned 4 × 4 (4.6 Å pixel⁻¹, 48-pixel box size). The particle stacks were subjected to reference-free 2D classification and non-particle picks were removed from the stacks. The resulting particle stacks, containing 2.5 M particles for dataset 1 and 1.0 M particles for dataset 2, were input to 3D auto-refinement. The initial model was obtained from a CryoSPARC ab initio reconstruction using a subset of the data. The refined stack for dataset 1 was input to 3D classification in RELION (4 classes, tau-value 4, 25 iterations, performing angular and translational searches). Particles belonging to a single high-resolution class, containing 1.1 M particles, were re-centered and re-extracted at a binning of 2 × 2 (2.3 Å pixel⁻¹, 96-pixel box size). Another round of 3D auto-refinement followed by 3D classification (4 classes, tau-value 4, 25 iterations, performing angular and translational searches) and selection of one

high-resolution class resulted in a stack of 288,860 particles, which were re-centered and re-extracted at full resolution (1.15 Å pixel⁻¹, 192-pixel box size). The refined particle stack for dataset 2 was re-centered and re-extracted at a binning of 2 × 2 (2.3 Å pixel⁻¹, 96-pixel box size) and subjected to 3D classification (4 classes, tau-value 4, performing angular and translational searches). A single high-resolution class was selected, containing 379,924 particles, re-centered and re-extracted at full resolution (1.15 Å pixel⁻¹, 192-pixel box size). Particles from each full resolution stack were grouped by beam shift and subjected to beam tilt refinement, prior to 3D auto-refinement. The full resolution stacks from datasets 1 and 2 were merged and 3D auto-refined to a resolution of 4.3 Å, with much improved quality of the EM density compared to reconstructions from the individual datasets. Inspection of the Euler distribution revealed preferred orientations and a lack of end-on views. First, the reconstruction was subjected to Euler normalization by capping the population in Euler space to a maximum of 3 standard deviations from the mean, resulting in removal of 58,799 particles from the stack. Second, missing views were added back into the stack as follows: Parallel processing of dataset 1 in CryoSPARC led to lower resolution reconstructions, however, end-on views were found in 2D classification. The coordinates of these end-on views were imported into RELION 3.1[65], particles were extracted at full resolution (1.15 Å pixel⁻¹, 192-pixel box size) and 2D classification followed by selection of particles showing secondary structure elements led to a stack containing 30,051 particles. The end-on particles were merged with the Euler normalized stack, no duplicate particles were found in the merged stack. Iterative CTF (first beam-tilt and trefoil refinement, followed by per-particle defocus, astigmatism, and B-factor refinement) and 3D auto-refinement resulted in a final reconstruction with a nominal resolution of 4.1 Å. The resolution of the reconstruction was not improved by Bayesian polishing. The local resolution was estimated using RELION[65] and the 3D Fourier Shell Correlation was calculated using the 3D FSC server[69].

### Human mSerRS-mtRNA^Ser(GCU)-TL cryo-EM data processing

Warp[62] was used for pre-processing of movies for mSerRS-mtRNA^Ser(GCU)-TL, the frames were aligned and dose weighted and CTF was estimated in 5 × 5 patches. A BoxNet model, trained on one representative micrograph, was used for particle picking and particles were extracted at full resolution (1.15 Å pixel⁻¹, 192-pixel box size). Next, the particle stack was imported into CryoSPARC[63] and subjected to 2D classification. Particles with well-defined 2D structural elements were selected, resulting in a particle stack containing 6.9 million particles. Ab initio reconstruction was performed, requesting 2 classes, of which one class complied with the expected size and shape of mSerRS-mtRNA^Ser(GCU)-TL. This class was homogeneously refined to a nominal resolution of 4.3 Å. Heterogeneous refinement, requesting 3 classes, showed one high-resolution class containing 1.7 million particles. This stack was subjected to homogeneous refinement, followed by a secondary heterogeneous refinement step, requesting 4 classes. Two high-resolution classes were selected and homogeneously refined to a nominal resolution of 4.1 Å. This stack of 118,269 particles was imported into RELION and subjected to 3D auto-refinement, which did not improve the reported resolution but did improve the quality of the EM density. CTF refinement with optic groups and Bayesian polishing did not yield further quantitative or qualitative improvements in resolution. The local resolution was estimated using RELION and the 3D Fourier Shell Correlation was calculated using the 3D FSC server[69].

### Atomic modeling and refinement of the cryo-EM structures

The crystal structure of mSerRS-SerSA complex (PDB 7TZB) and the *E. coli* tRNA^Phe structure (PDB 3L0U) were used as starting models and were rigid body fit into the EM density maps. The models for the

tRNA were trimmed to appropriate chain lengths and bases were renumbered appropriately. The tRNA models were rebuilt in regions that deviated substantially from the *E. coli* tRNA^Phe structure, specifically for the anticodon stem and core regions of mtRNA^Ser(GCU) and mtRNA^Ser(GCU)-TL. The geometry of the tRNA models was optimized using ERRASER[70] through the Rosetta Online Server (https://rosie.graylab.jhu.edu/). The models were real-space refined in Coot, restrained to ideal geometry, secondary structure, and Geman-McClure distance restraints generated in ProSMART[71] from the input models. The models were iteratively real-space refined in Coot and in Phenix-1.20.1 (by rigid body and global minimization) using Ramachandran and secondary structural restraints. The model was further optimized for compliance to geometric constraints using MolProbity[72] as guidance and by geometry minimization in Phenix-1.20.1. MolProbity was used to assess the quality of the final model and report validation statistics in Supplementary Table 2.

### Figure generation
Figures were rendered in ChimeraX[73].

### Quantification and statistical analysis
The statistical analysis of the X-ray crystallographic and cryo-EM data processing, model building, and model refinement is described in Methods details and summarized in Supplementary Tables 1 and 2.

### Reporting summary
Further information on research design is available in the Nature Research Reporting Summary linked to this article.

### Data availability
All data generated or analyzed during this study are included in this published article (and its supplementary information files) and are available for the corresponding author upon request. Cryo-EM maps of mSerRS-mtRNA^Ser(GCU) and mSerRS-mtRNA^Ser(GCU)-TL have been deposited in the Electron Microscopy Data Bank (EMDB) under the accession codes EMD-26310 and EMD-26311, respectively. Atomic coordinates of the models have been deposited in the Protein Data Bank (PDB) under accession codes 7U2A, 7U2B, and 7TZB for the crystal structure of mSerRS bound to SerSA. The atomic coordinates used for molecular replacement or structural comparison were obtained from the Protein Data Bank: 1SER, 1U2A, 1WLE, 3L0U, 4RDX, 4RQF, 4TRA, and 6UGG. All tRNA gene sequences were retrieved from the tRNAdb/mitotRNAdb (http://trna.bioinf.uni-leipzig.de/) and genomic tRNA database (GtRNAdb; http://gtrnadb.ucsc.edu/). Source data are provided with this paper.

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

## Acknowledgements

We thank Sergey Melnikov, Venki Ramakrishnan, and Brian Wimberly for critical reading of earlier versions of the manuscript and valuable suggestions. We thank Ingrid Vallee for help with the acquisition of mass photometry data and Claudio Zambaldo for help with the synthesis of SerSA. X-ray crystallographic experiments were carried out at Stanford Synchrotron Radiation Lightsource (SSRL). P.S. and B.K. are supported by the National Institutes of Health (NIH) Grant R01 GM125908. P.S. is supported by a fellowship from the National Foundation for Cancer Research, and a fellowship from the Skaggs Foundation. G.C.L. is supported by the National Institutes of Health (NIH) AG067594 and AG061697. L.K.D. is supported by the Human Frontier Science Foundation. Computational analyses of EM data were performed using shared instrumentation funded by NIH S10OD021634 to G.C.L.

## Author contributions

B.K. conceived the study, designed research, performed biochemical experiments, prepared samples for structural experiments, performed X-ray crystallographic experiments, conducted model building, analyzed the data, and wrote the manuscript with input from all authors. M.H. conducted the electron microscopy experiments, single-particle 3D reconstruction, and model building under the guidance of G.C.L. L.K.D. and P.S. provided conceptual and mechanistic insights and co-wrote the manuscript.

## Competing interests

The authors declare no competing interests.
