## [Peer Review File · Nature Communications]

Structural basis for shape-selective recognition and aminoacylation of a D-armless human mitochondrial tRNAReviewers' Comments:

Reviewer #1:

Remarks to the Author:

The article titled "Structural basis for shape-selective recognition and aminoacylation of a D-armless human mitochondrial tRNA" by Kuhle et al., has utilized cryo-electron microscopy to elucidate the mechanism of unusual mtRNA selection by mSerRS from *Homo sapiens*. As an extension of their previous work (Kuhle et al. *Nat Commun*, 2020), the authors have explained how the loss of identity elements from mtRNASer in bilaterians allowed mSerRS to evolve a unique tRNA capturing mechanism for mitochondrial protein synthesis. The work has elegantly demonstrated how SerRS recognizes the non-canonical tRNASer which has lost the variable loop and D-arm. The authors have used cryoEM with a unique particle picking strategy to solve preferred orientation problem and provided the first structural insights into a degenerate tRNA binding to aaRS. A key finding of the study is that the mtRNASer adopts a unique and flexible tertiary structure and the mSerRS utilizes the altered T-arm topology for serylation. The structural work is well done and the manuscript is written up clearly. There are some concerns, listed below, which the authors can address in a revised manuscript.

Major points:

- It is mentioned, in line number 231-233, that this structure provides a mechanistic basis to understand the principles of co-evolution between nuclear-encoded proteins and the rapidly evolving mitochondrially encoded tRNA substrates. How many tRNAs are degenerated in mitochondria? How can this anticodon independent aaRS recognition be applied to all degenerate aaRS-tRNA pairs? These points need to be elaborated.
- Animal mitochondrial translation systems contain two serine tRNAs, each possessing an unusual secondary structure; tRNAGCUSER lacks the entire D arm, whereas tRNAUGASER (for UCN) has a slightly different cloverleaf structure. Both are efficiently charged by mitochondrial SerRS. How this structure and shape-selective mechanism explains the SerRS recognition of both tRNAs?
- Considering the relatively small size of the complex, coupled with high-angle tilting (40 deg) during data collection, how much of the high angle tilt data contributed to the final reconstruction. Could the authors have coupled tilted angle data collection with Euler normalization to improve the angular distribution of particles? A histogram of the number of particles from each of the tilts that went into the final reconstruction will be informative.
- Authors have shown in their previous article (Kuhle et al, *Nat Com*, 2020) that increasing the stability of tRNA^{Asp} by mutating few residues in both D- and T-loop results in its charging by mitochondrial SerRS. However, degenerate tRNASer will have lower stability as seen by flexibility in anticodon loop. The recognition of degenerate tRNASer by SerRS seems in contrast to what was proposed earlier.
- Authors state that they have addressed how the long-term functional decline of mitochondrial gene expression prevented against the continuous accumulation of mutations in its mtRNA genes? However, they present no data in support of that.
- From line 274 to 288, the authors propose that genetic drift and mutations are responsible for mito-tRNA changes that may cause diseases, but they also suggest that the mito-tRNA changes have an evolutionary advantage?? Authors should clarify these statements.

Minor points:

- Following information is missing from the supplementary table 1: Diffraction source, X-ray wavelength, total reflections, unique reflections, Wilson B-factor, reflection used for Rfree, average B-factor for protein residues and ligands, and Molprobit score. R should be italics in space group (R3) name.
- In line number 265: "its" to "their"!
- In line number 327: authors should mention how and why tRNAs were cleaved before gel purification.
- In line number 200: Missing reference to Fig. 4e (4d?)!

- In line number 70: sequence and structural similarity between human mSerRS and its bovine counterpart needs to be provided.
- In line number 147: "virtually invariant"???
- In Pg.20, Lines 411-412, the sentence "Cryo-EM grids were prepared by application of 4 μ L sample, followed by manual blotting using Whatman No. 1 filter paper for 4-5 seconds and plunge freezing in liquid ethane at 4 °C in 95% humidity" sounds confusing!

Rajan Sankaranarayanan

Reviewer #2:

Remarks to the Author:

Regarding the ms "Structural basis for shape-selective recognition and aminoacylation of a D-armless human mitochondrial tRNA" by Kuhle et al.

Mammalian mitochondria use two tRNA(Ser) species, one for AGY codons and one for UCN codons. Both species are equally well recognized and charged by the mitochondrial seryl-tRNA synthetase (termed mSerRS in the manuscript). Both tRNA(Ser(AGY)) and tRNA(Ser(UCN)) display non-canonical tertiary structures based on structure predictions, NMR-data and RNase probing data. This is an important study both in relation to tRNA evolution in general but also for the field of mitochondrial gene expression.

In the ms, Kuhle et al., determine the crystal structure of unbound mSerRS by macromolecular crystallography (MX) and cryo-EM structures of complexes between one tRNA (Ser(AGY)) bound to the dimeric mSerRS. Although the structural work is solid, figures and graphs well presented, I find the manuscript disappointing in several aspects:

1. The binding between the single tRNA(Ser(AGY)) and mSerS is analyzed in detail but there is no discussion on how the second substrate tRNA(Ser(UCN)) in human mitochondria may be bound to mSerS. We know from previous work that the two different tRNA substrates appear to have a different binding mode (Shimada et al. 2001 for the human system) and the extensive mutagenesis work in Chimnarok et al., 2005 for the closely related bovine system. I don't even find it even mentioned anywhere in the ms that tRNA(Ser(UCN)) is the second substrate of mSerS but do not find it meaningful to speculate on the motivation for this glaring omission and the lack of corresponding analysis/discussion. This should be addressed in the ms along with a result/discussion section based on the mutagenesis work (esp. differences in the effects on charging for the two different tRNA substrates) in Chimnarok et al., 2005 is warranted.
2. What is analyzed in detail is the structural differences between tRNA(Ser(AGY)) as bound to mSerRS to Yeast tRNA(Phe). This is a standard comparison and competently done but the weakness from the very outset is that this is the structure of tRNA(Ser(AGY)) when bound to mSerRS not at all necessarily the structure of the tRNA in solution so wider and longer evolutionary speculations should be toned down. Furthermore, 48-50 in tRNA(Ser(AGY)) are m5C-modified in human mitochondria and this will affect the local structure and interactions in this region close to the mSerRS-tRNA(Ser(AGY)) binding interface thereby warranting some caution in the resulting conclusions from the present mSerRS-tRNA(Ser(AGY)) structure using unmodified tRNA.
3. The work presents the structure of the dimeric mSerRS bound to one tRNA but previous work (e.g. Shimada et al., 2001 for the human system) shows that two tRNAs are bound to the mSerRS dimer. An active site titration assay is described in the methods (p.16-17) but as far as I can see there are no results reported from this assay. It is further mentioned that there are 1.5 times more tRNA than protein added for the gel-filtration isolation of the complex but given the relative instability and folding issues of tRNA(Ser(AGY)) this may not be enough to saturate both binding sites in the dimer. Given the results of previous work, I would have wanted to see both the results of the active site titration assay performed but not reported and an analytical GF elution profile with at least 5-10 time molar

excess of tRNA(Ser(AGY)) to be convinced that the true interaction mode between mSerRS and tRNA(Ser(AGY)) is indeed 2:1. Furthermore, if it is indeed the case that only one tRNA binds to the mSerS dimer at a time, one would want to understand the molecular basis for the hindrance of the binding of a second tRNA to the dimer once a first tRNA is bound.

4. The resolution and quality of the cryo-EM reconstructions are disappointing given modern standards. Authors should investigate if for example Bayesian polishing combined with refinement of global aberrations (divided for beam/image-shift groups from the Legimon collection) can improve the resolution of the reconstructions especially since the ms describes and delineates important molecular aspects of tRNA structure and tRNA/protein interactions.

Minor comments:

5. Suppl. Table 1: R3 implies a hexagonal setting in MX so stating both a and b lengths and unit cell angles is superfluous. Version of Phenix used should be stated both in the table and in the methods text. Should be stated/noted that it is the average B-factor and please separate protein and ligand. For Ramachandran, rotamer and clash score statistics it should be stated if it is according to Molprobit or some other validation program.

6. Suppl. Table 2: Typo for the unit in electron exposure. Version of Phenix used should be stated both in the table and in the methods text. It should be stated/noted that it is the average B-factor and please separate protein and tRNA. For Ramachandran, rotamer and clash score statistics it should be stated if it is according to Molprobit or some other validation program.

7. p. 9 "the" missing in parts in the descriptions relating to the N-helix and the C-tail.

8. In the methods section, it is not detailed which pyrophosphatase is used (E. coli, yeast or..).

9. p. 20, UltraAuFoil is from Quantifoil, EMS is a distributor.

10. p. 21 the initial model seems to have been generated twice, first from Warp-picked particles and second from LOG-picked particles? Please clarify.

11. p. 21 "Iterative CTF and 3D auto refinement resulted". Here it is unclear if it is just the local defoci that are refined or also global aberrations? If also global aberrations, which of them and in which order?

12. p. 22 For the TL-variant tRNA structure: It is unclear if local defoci and global aberrations (which of them and in which order) are refined either in CryoSPARC or after export to the RELION suite.

13. Suppl. Figures 5,6C. The resolution range chosen is strange. Surely there must be parts of the reconstruction that are better than 3.9-4 Å? A range of 3-6 or 3-5 Å would be more informative.

14. RELION is used but not referenced.

15. The names of the software suite RELION should be capitalized throughout the ms. Please also capitalize XDS in ref. 50. Furthermore, Phenix is capitalized in the MX methods part but not in the cryo-EM part so please adjust for consistency (Phenix is preferred by the Phenix authors).

16. Suppl. Fig. 2B. Choose a thinner C- α representation than cartoon and choose a contrasting color scheme for overall structural comparison of the bovine and human mSerRS.

17. MolProbit should have a capital P.

Reviewer #3:

Remarks to the Author:

Review of Kuhle et al

This manuscript is bursting with exciting new data and insights. The authors have solved the structure of a complex between the human mitochondrial seryl-tRNA synthetase (mSerRS) and its cognate mitochondrial tRNA^{Ser}. The work is technically state of the art, and the conclusions are quite striking, documenting one of the more penetrating and interesting aspects of the structural biology of genetic coding to emerge in recent years.

Mitochondrial tRNAs have long been of considerable interest because they exhibit substantially more violations of the canonical "L-shaped" structure than do tRNAs from any other domain, in spite of the fact that the cognate aminoacyl-tRNA synthetases (aaRS), generally encoded in the nucleus, have almost exclusively canonically bacterial sequences and structures. Mitochondrial tRNAs, encoded by the mitochondrial genome, are often deficient in one or several of the features thought to be canonical, and which often include what are described as "identity" elements because they help define the specific recognition interfaces with cognate aaRS. Of the variant mitochondrial tRNAs, mitochondrial seryl-tRNA, mtRNASer, deviates the most from canonical tRNAs, because it lacks the entire dihydrouridine stem loop, which contributes with the T ψ C loop to form the tertiary base pairs that form the core stabilizing the L shape. Kuhle et al. have used X-ray crystallography to solve the structure of the mitochondrial seryl-tRNA synthetase, mSerRS, in the absence of its cognate mtRNASer, and Cryo electron microscopy to solve its complexes with several mtRNASer variants.

The structures reveal quite surprising consequences of the loss of the D-stem loop. The complex most closely resembles that of the corresponding bacterial complex in the interaction of the acceptor stem and T ψ C stem-loop, which at first glance appear quite homologous. The details, however, reveal quite significant changes in both the mtRNASer structure and complementary differences in the mSerRS conformation. Most surprising is that in the absence of the canonical core tertiary base pairs, and perhaps in the absence of the long variable arm, the anticodon stem loop is untethered and exhibits a highly flexible structure that cannot be completely specified in any of the electron density maps, but which appears from the complex with a stabilized variant of mtRNASer to form an acute angle that is aptly described as forming a "Y" shaped structure, rather than the canonical "L"-shape.

The extraordinary biological relevance of these differences is that nearly all of the canonical "Identity" elements that inform the specific recognition between aaRS and tRNA have been replaced by much more base-sequence tolerant secondary structural features. The most evident of these is that the T-stem is one base-pair longer than the canonical mtRNASer, which rotates and re-models the T-loop, allowing the two bases extruded in canonical tRNAs to form tertiary base triples with the (missing) D-loop to base pair instead with the 3' antiparallel strand of the T-loop. The long α -helical N-terminal helical coiled-coil domain curves more tightly around this re-configured T-loop, substantially increasing the surface area buried between that domain and the minor groove of the elongated T-stem. That the essential contacts in the complex involve only the minihelix is remarkably consistent with Professor Schimmel's earlier suggestion that cognate interactions might have developed first between tRNA-like minihelices before the advent of the D- and anticodon-stem loops (Schimmel, et al., Proc. Nat. Acad. Sci. USA, 1993, 90, 8763-8768).

Acceptor stem interactions are more canonical, but even these are altered to increase the base sequence independence of the cognate interaction. The mSerRS motif 2 loop is shortened, precluding interactions that confer sequence-preferences in positions 2:71 and 3:70 by the bacterial SerRS. Those interactions are replaced instead by backbone amide hydrogen bonds. In this respect, as with the T stem-loop interaction, the changes tend to eliminate base sequence dependence of specific recognition.

These details and others are outlined clearly and without ambiguity, anticipating and answering adequately all the questions I might have raised from the structure and making the manuscript a joy to read. Moreover, the methods section describes a number of technically advantageous high-throughput modifications of routinely used methods as well as for synthesis of Ser-5' sulfamoyl adenylate.

The only comment I might add is that the new structural details also have rather profound evolutionary implications, not only for the (relatively) recent evolutionary divergence of mitochondrial translation machinery, which the authors describe, but also in an opposite sense, for the early ancestry of quite early aaRS•tRNA cognate pairs. Acceptor-stem interactions with the motif 2 loop illustrate key elements of what differentiate Class II aaRS tRNA substrates from those for Class I

tRNAs, namely the fully extended 3'DCCA. Further, dependence on primary sequence tolerant mechanisms such as structural shape rather than base sequence recognition were also probably key to the early formation of synthetase•tRNA cognate pairs (C. W. Carter, Jr and P. R. Wills, NAR, 2018, 46, 9667–9683; C. W. Carter, Jr and P. R. Wills, IUBMB Life, 2019, 71, 1088–1098).

Minor criticisms:

A trivial, but important source of confusion occurs on line 74 of page 4. The α/β terminology is at variance with long-standing description of α/β proteins as those such as TIM barrel and Rossmann dinucleotide binding proteins, which are dominated by parallel β -strands, whereas Class II aaRS are primarily antiparallel β structures in which the alpha helices are segregated from the beta strands. Thus, this is a confusing nomenclature, violating a classification introduced by Levitt and Chothia in 1976, and which has remained in continuous use ever since. According to those authors, the segregation of strand and antiparallel strand structures in Class II aaRS is termed $\alpha+\beta$. On line 111, page 6, the authors might remind readers that even canonical SerRS•tRNA^{Ser} complexes exhibit no specific recognition of the anticodon stem-loop because there are six serine codons two of which have a different middle base from the other four.

REVIEWER COMMENTS

Reviewer #1 (Remarks to the Author):

The article titled “Structural basis for shape-selective recognition and aminoacylation of a D-armless human mitochondrial tRNA” by Kuhle et al., has utilized cryo-electron microscopy to elucidate the mechanism of unusual mtRNA selection by mSerRS from Homo sapiens. As an extension of their previous work (Kuhle et.al. Nat commun, 2020), the authors have explained how the loss of identity elements from mtRNASer in bilaterians allowed mSerRS to evolve a unique tRNA capturing mechanism for mitochondrial protein synthesis. The work has elegantly demonstrated how SerRS recognizes the non-canonical tRNASer which has lost the variable loop and D-arm. The authors have used cryoEM with a unique particle picking strategy to solve preferred orientation problem and provided the first structural insights into a degenerate tRNA binding to aaRS. A key finding of the study is that the mtRNASer adopts a unique and flexible tertiary structure and the mSerRS utilizes the altered T-arm topology for serylation. The structural work is well done and the manuscript is written up clearly. There are some concerns, listed below, which the authors can address in a revised manuscript.

We thank the reviewer for his positive feedback and constructive criticism.

Major points:

- It is mentioned, in line number 231-233, that this structure provides a mechanistic basis to understand the principles of co-evolution between nuclear-encoded proteins and the rapidly evolving mitochondrially encoded tRNA substrates. How many tRNAs are degenerated in mitochondria? How can this anticodon independent aaRS recognition be applied to all degenerate aaRS-tRNA pairs? These points need to be elaborated.

As mentioned in the introduction (from line 48) and discussed (from line 284), all animal mtRNAs are subject to the same degenerative forces of high mutation rates and weak selection. Although the degree of degeneration varies between the 22 animal mtRNAs, nearly all of them (with exception of mtRNA^{Gln}) experienced strong degeneration of canonical structure and sequence elements. In all systems tested so far, this resulted in the inability of bacterial aaRSs to charge their corresponding “cognate” mtRNAs, regardless of whether they were anticodon-dependent or independent. This suggests that the evolutionary forces that led to the rewiring of the mitochondrial serylation system apply to other mitochondrial aaRS-tRNA pairs as well, including those that rely primarily on anticodon recognition.

As we discuss from line 275 onward, the mitochondrial aminoacylation systems, in particular the tyrosylation, aspartylation, and alanylation systems, and others underwent similarly fundamental shifts, away from sequence-specific recognition in favor of a more shape- and structure-based recognition mechanism. Thus, although the mechanistic details may differ, our analysis of the altered mSerRS-mtRNA interaction provides molecular insight into a coevolutionary process that most if not all mitochondrial aaRS-tRNA pairs are subject to.

- Animal mitochondrial translation systems contain two serine tRNAs, each possessing an unusual secondary structure; tRNAGC^{Ser} lacks the entire D arm, whereas tRNAUGA^{Ser} (for UCN) has a slightly different cloverleaf structure. Both are efficiently charged by mitochondrial SerRS. How this structure and shape-selective mechanism explains the SerRS recognition of both tRNAs?

Thank you for bringing this up. Indeed, the two animal mtRNA^{Ser} isoacceptors do not share the same secondary and tertiary structure. Importantly, they also do not share any conserved sequence elements that could be used for a common sequence-specific recognition mechanism. Earlier biochemical analyses of the bovine mitochondrial serylation system suggested that both mtRNA^{Ser} isoacceptors are recognized in the elbow region, despite their lack of structural homology [Shimada et al, JBC, 2001; Chimnarok et al, EMBO J, 2005]. This would suggest that mSerRS uses two distinct interfaces and two distinct binding modes to recognize its two mtRNA^{Ser} substrates. The molecular details of how this dual recognition is achieved remain an important open question. To address this point, we included a paragraph into the discussion section (from line 300) of the revised manuscript.

- Considering the relatively small size of the complex, coupled with high-angle tilting (40 deg) during data collection, how much of the high angle tilt data contributed to the final reconstruction. Could the authors have coupled tilted angle data collection with Euler normalization to improve the angular distribution of particles? A histogram of the number of particles from each of the tilts that went into the final reconstruction will be informative.

We initially performed Euler normalization on the untilted particle stack in an attempt to obtain a reconstruction with more isotropic resolution, however artifacts remained due to the prevalence a single view. We performed the same Euler normalization on the combined stack of untilted and tilted particles, but the quality of the reconstruction decreased so that modeling became challenging. Although there remain negligible artifacts from preferred views, we feel the overall interpretable quality of the reconstruction nonetheless benefits from inclusion of these views into the back projection.

The number of particles from each of the tilts that went into the final reconstruction are summarized in the following table. Since the number of movies collected at the various tilt angles varied, we included the average number of particles per movie for each of the tilt angles.

Tilt angle	# movies	# final particles	Average # final particles/movie	% particle contribution
0	64	12916	202	11%
20	346	31201	90	26%
40	920	74152	81	63%

- Authors have shown in their previous article (Kuhle et al, Nat Com, 2020) that increasing the stability of tRNA^{Asp} by mutating few residues in both D- and T-loop results in its charging by mitochondrial SerRS. However, degenerate tRNA^{Ser} will have lower stability as seen by flexibility in anticodon loop. The recognition of degenerate tRNA^{Ser} by SerRS seems in contrast to what was proposed earlier.

The answer to this question is related to the reviewer's question above regarding the second mtRNA^{Ser(UGA)} isoacceptor. From what we know about mtRNA^{Ser(UGA)} from earlier studies [Watanabe et al, NAR, 1994; Hayashi et al, JMB, 1998], its elbow region is near-canonical, including the G19:C56 Watson-Crick base-pair between D-loop and T-loop. According to biochemical data from the bovine system [Shimada et al, JBC, 2001; Chimnaronk et al, EMBO J, 2005], this tertiary interaction (or the integrity of the elbow region in general) is important for mtRNA^{Ser(UGA)} recognition by bovine mSerRS (but not for mtRNA^{Ser(GCU)}). What we did in our earlier work [Kuhle et al, Nat Com, 2020] was essentially to restore in mtRNA^{Asp} the canonical stable D-loop-T-loop interactions observed in mtRNA^{Ser(UGA)}, thereby removing the aminoacylation barrier within the non-cognate mSerRS-mtRNA^{Asp} pair. The apparent contrast between these data and our current proposal arises from the apparent bimodality of tRNA recognition by mSerRS described above: On one hand the recognition of mtRNA^{Ser(GCU)} through its remodeled T-loop structure. On the other the recognition of mtRNA^{Ser(UGA)}, presumably through the canonical elbow structure [Shimada et al, JBC, 2001]. Our previous article [Kuhle et al, Nat Com, 2020] 'made use' of the latter recognition mode to demonstrate how the evolutionary loss of canonical structural features in nearly all mtRNAs contributes to the maintenance of aminoacylation barriers in human mitochondrial translation.

- Authors state that they have addressed how the long-term functional decline of mitochondrial gene expression prevented against the continuous accumulation of mutations in its mtRNA genes? However, they present no data in support of that.

We thank the reviewer for bringing this up. In lines 58-60 of the original manuscript, we asked: "And how is the long-term functional decline of mitochondrial gene expression prevented against the continuous accumulation of mutations in its mtRNA genes?". To clarify, we did not intend to ask "...how **does** the long-term functional decline of mitochondrial gene expression prevented against the continuous accumulation of mutations...". We recognize that the original wording of this passage is misleading.

What we meant is the following: The continuous accumulation of mutations erodes sequence and structure elements in mtRNAs. Corresponding mutational changes in canonical gene expression machineries would inevitably result in a decline or even complete loss of function (which is why such mutations are effectively purged from the population by purifying selection). **How can mitochondrial proteins, in particular mt-aaRSs, and mtRNAs maintain their critical molecular function despite the continuous mutational change in the latter?**

Our analysis of the mSerRS-mtRNA^{Ser(GCU)} interaction addresses this question by showing that 1.) their intermolecular interactions were fundamentally rewired in response to the loss of canonical identity elements in the mtRNA, and 2.) that these interactions became less dependent on conserved sequence elements, thus allowing more sites in mtRNAs to be mutated without loss of function.

To improve its clarity, we rephrased the passage (line 57 onwards). It now reads: "...and it remains unclear how the long-term functional decline of mitochondrial gene expression is prevented, despite the continuous accumulation of mutations in mtRNA genes."

- From line 274 to 288, the authors propose that genetic drift and mutations are responsible for mito-tRNA changes that may cause diseases, but they also suggest that the mito-tRNA changes have an evolutionary advantage?? Authors should clarify these statements.

We thank the reviewer for pointing out this potential source of confusion. There is no doubt that mutation accumulation by genetic drift and Muller's ratchet is one if not the main driving force for evolutionary change in mtRNAs. The currently unresolved question is, however, whether genetic drift is the only force, or whether adaptation and selection also play a role in shaping the unusual structures of extant animal mtRNAs. In the first case, all the unusual structural features would be exclusively the result of mutational erosion. In the second case, a **subset** of these unusual structural features would be maintained by natural selection. Our results suggest that the latter is the case. To clarify: we do not suggest that these changes confer an evolutionary **advantage** (a phrase that we do not use in the manuscript) in the sense that they **improve** the biophysical properties of the mtRNA or confer higher individual fitness to the organism. Instead, we suggest that changes such as the remodeled T-loop of mtRNA^{Ser(GCU)}, which serves as idiosyncratic recognition element for mSerRS, help the mitochondrial gene expression machinery to **maintain** its function under conditions of elevated mutation pressure. In that sense, such changes confer an 'advantage' only by making the mitochondrial translation machinery less susceptible to long-term functional decline by mutational meltdown. At the same time, however, they are unable to prevent the general mutational degeneration and destabilization of mtRNAs that are responsible for the increased susceptibility of human mtRNAs to disease-causing mutations. We hope that this clarifies the concern raised by the reviewer.

Minor points:

- Following information is missing from the supplementary table 1: Diffraction source, X-ray wavelength, total reflections, unique reflections, Wilson B-factor, reflection used for Rfree, average B-factor for protein residues and ligands, and Molprobit score. R should be italics in space group (R3) name.

We thank the reviewer for catching the missing information, which was added in the revised Supplementary Table 1.

- In line number 265: "its" to "their"!

The passage was corrected and now reads as follows (line 275): "The identity set of the human mitochondrial recognition system shows virtually no overlap with its ancestral bacterial SerRS-tRNA^{Ser} counterpart."

- In line number 327: authors should mention how and why tRNAs were cleaved before gel purification.

We thank the reviewer for noting this. There was no cleavage step in the purification of the tRNA. The passage was changed accordingly (line 343).

- In line number 200: Missing reference to Fig. 4e (4d)?!

Thank you for noting this. The missing reference was included (line 209).

- In line number 70: sequence and structural similarity between human mSerRS and its bovine counterpart needs to be provided.

The r.m.s. deviation for the two structures was provided in the legend of the referenced Supplementary Figure 2b. We included the sequence identity in the same legend. We hope that the reviewer agrees with our decision to not include it in the main text.

- In line number 147: “virtually invariant”???

We changed “invariant” to “unchanged” (line 154).

- In Pg.20, Lines 411-412, the sentence "Cryo-EM grids were prepared by application of 4 μ L sample, followed by manual blotting using Whatman No. 1 filter paper for 4-5 seconds and plunge freezing in liquid ethane at 4 °C in 95% humidity" sounds confusing!

We thank the reviewer for this suggestion to improve clarity in the language. The confusing sentence is changed to the following (line 433):

“Cryo-EM grids were prepared by application of 4 μ L protein sample at 4 °C in 95% humidity. The grids were manually blotted for 4-5 seconds using Whatman No. 1 filter paper, followed by plunge freezing in liquid ethane.”

Rajan Sankaranarayanan

Reviewer #2 (Remarks to the Author):

Regarding the ms “Structural basis for shape-selective recognition and aminoacylation of a D-armless human mitochondrial tRNA” by Kuhle et al.

Mammalian mitochondria use two tRNA(Ser) species, one for AGY codons and one for UCN codons. Both species are equally well recognized and charged by the mitochondrial seryl-tRNA synthetase (termed mSerRS in the manuscript). Both tRNA(Ser(AGY)) and tRNA(Ser(UCN)) display non-canonical tertiary structures based on structure predictions, NMR-data and RNase probing data. This is an important study both in relation to tRNA evolution in general but also for the field of mitochondrial gene expression.

We are glad that the reviewer considers our work to be important and thank her/him for the constructive criticisms and comments.

In the ms, Kuhle et al., determine the crystal structure of unbound mSerRS by macromolecular crystallography (MX) and cryo-EM structures of complexes between one tRNA (Ser(AGY)) bound to the dimeric mSerRS. Although the structural work is solid, figures and graphs well presented, I find the manuscript disappointing in several aspects:

1. The binding between the single tRNA(Ser(AGY)) and mSerS is analyzed in detail but there is no discussion on how the second substrate tRNA(Ser(UCN)) in human mitochondria may be bound to mSerS. We know from previous work that the two different tRNA substrates appear to have a different binding mode (Shimada et al. 2001 for the human system) and the extensive mutagenesis work in Chimnarok et al., 2005 for the closely related bovine system. I don't even find it even mentioned anywhere in the ms that tRNA(Ser(UCN)) is the second substrate of mSerS but do not find it meaningful to speculate on the motivation for this glaring omission and the lack of corresponding analysis/discussion. This should be addressed in the ms along with a result/discussion section based on the mutagenesis work (esp. differences in the effects on charging for the two different tRNA substrates) in Chimnarok et al., 2005 is warranted.

We thank the reviewer for raising this point. To clarify: Our motivation to keep mtRNA^{Ser(UGA)} out of the analysis/discussion was to strictly maintain the focus on our main goals: 1.) To determine a structure of the shortest and most degenerated human mitochondrial tRNA (i.e. mtRNA^{Ser(GCU)}), and 2.) to understand its molecular recognition by the cognate synthetase. The mSerRS-mtRNA^{Ser(GCU)} structure is not only the first animal mitochondrial aaRS-tRNA complex structure to be determined so far, but also reveals unprecedented structural deviations in mtRNA^{Ser(GCU)} and its interactions with mSerRS. We are therefore convinced that our analysis should maintain its original focus on the unique molecular properties of this complex and its functional and evolutionary implications.

While we agree that it would be very interesting to understand the molecular basis for the recognition of mtRNA^{Ser(UGA)} and the dual recognition mechanism of both mtRNA^{Ser}, a detailed

comparison, analysis, and discussion would in our opinion confound the important results and clarity of this manuscript. This would be especially true as we do not provide the corresponding structure of the human mSerRS-mtRNA^{Ser(UGA)} complex as a basis for a balanced comparison and discussion. We therefore hope that the reviewer agrees with our discussion that a full analysis of the dual recognition of both isoacceptors goes beyond the intended scope and goals of this manuscript.

We included a paragraph regarding mtRNA^{Ser(UGA)} and the problem of dual recognition of both mtRNA^{Ser} isoacceptors into the discussion of the revised manuscript (from line 300).

2. What is analyzed in detail is the structural differences between tRNA(Ser(AGY)) as bound to mSerRS to Yeast tRNA(Phe). This is a standard comparison and competently done but the weakness from the very outset is that this is the structure of tRNA(Ser(AGY)) when bound to mSerRS not at all necessarily the structure of the tRNA in solution so wider and longer evolutionary speculations should be toned down.

If we understand correctly, the reviewer suggests that mtRNA^{Ser(GCU)} may undergo a larger conformational transition between its free and mSerRS-bound state. Since we discussed in our manuscript the possibility that mSerRS-binding favors particular conformational states of mtRNA^{Ser(GCU)} relative to its free state by reducing its intrinsic conformational flexibility (from line 114), we assume the reviewer refers to the possibility that 1.) the T-loop of mtRNA^{Ser(GCU)} may adopt a different structure in solution or 2.) that the free tRNA may form tertiary interactions between T-loop and D-replacement loop. Although we agree that we cannot exclude the possibility of local changes in the T-loop, e.g. by its destabilization, the available evidence does not support major conformational rearrangements into a stable alternative structure. Ueda et al. have shown that, consistent with the structure of mSerRS-bound human mtRNA^{Ser(GCU)}, the T-loop of bovine mtRNA^{Ser(GCU)} is particularly sensitive to single-stranded RNA-specific S1 nuclease cleavage in solution (at positions U55 and U58 (C58 in humans)), similar to the tRNA's anticodon [Ueda et al, J Biochem, 1985], suggesting the absence of stable protective tertiary interactions. Moreover, earlier data for the human and bovine mtRNA^{Ser(GCU)} in solution suggest that these tRNAs lack a stable tertiary core and instead show increased flexibility between acceptor and anticodon domains [Frazer-Abel & Hagerman, NAR, 2008]. Together with the fact that conserved canonical sequence motifs are absent, there is currently no evidence to suggest that mtRNA^{Ser(GCU)} adopts an altogether different or even a more canonical tertiary structure in solution. We rephrased our discussion from line 251 to address this point.

Finally, we are not sure which of the evolutionary implications would be affected, should the solution structure of mtRNA^{Ser(GCU)} indeed differ in some respects from the one found in our complex structure, especially since our discussion of the evolutionary implications is focused almost exclusively on the interactions of mtRNA^{Ser(GCU)} with the synthetase and not on its structure in solution. We hope the reviewer agrees with our clarification of this point.

Furthermore, 48-50 in tRNA(Ser(AGY)) are m⁵C-modified in human mitochondria and this will affect the local structure and interactions in this region close to the mSerRS–tRNA(Ser(AGY)) binding interface thereby warranting some caution in the resulting conclusions from the present mSerRS-tRNA(Ser(AGY)) structure using unmodified tRNA.

Thank you for raising this point. Indeed, human mtRNA^{Ser(GCU)}, like many cytoplasmic tRNAs, is m⁵C modified in positions 48, 49, and 50 [Suzuki et al, Nat. Com., 2020]. While the precise function of these modifications remains unknown, they were suggested in cytoplasmic tRNAs to stabilize the stacking and local structure of the V-loop-T-arm junction and to prevent endonucleolytic cleavage. As suggested by the reviewer, we included a discussion of the possible structural and functional consequences of these m⁵C modifications in the revised manuscript (lines 117 and 203, and the legend to Fig. 4c).

3. The work presents the structure of the dimeric mSerRS bound to one tRNA but previous work (e.g. Shimada et al., 2001 for the human system) shows that two tRNAs are bound to the mSerRS dimer. An active site titration assay is described in the methods (p.16-17) but as far as I can see there are no results reported from this assay. It is further mentioned that there are 1.5 times more tRNA than protein added for the gel-filtration isolation of the complex but given the relative instability and folding issues of tRNA(Ser(AGY)) this may not be enough to saturate both binding sites in the dimer. Given the results of previous work, I would have wanted to see both the results of the active site titration assay performed but not reported and an analytical GF elution profile with at least 5-10 time molar excess of tRNA(Ser(AGY)) to be convinced that the true interaction mode between mSerRS and tRNA(Ser(AGY)) is indeed 2:1. Furthermore, if it is indeed the case that only one tRNA binds to the mSerS dimer at a time, one would want to understand the molecular basis for the hindrance of the binding of a second tRNA to the dimer once a first tRNA is bound.

Thank you for raising this point. We indeed tested a higher excess of tRNA over mSerRS (up to 5-fold), which, however, did not result in an increased occupancy of the mSerRS dimer with tRNA. Below we show the overlay of chromatograms for mSerRS run on a SEC column with 1.5 and 5-fold excess of mtRNA^{Ser(GCU)}, respectively. In both cases, only one major peak is formed for the mSerRS–mtRNA^{Ser(GCU)} complex, which elutes at the same volume for both protein:tRNA ratios. This peak was used subsequently for structure determination, showing the reported 2:1 ratio. Unfortunately, SEC does not provide sufficiently high resolution to confidently determine whether the complex, when eluting from the column, contains one or two tRNAs. Thus, we cannot rule out the possibility that the final 2:1 ratio is the result of a partial dissociation of the complex following its isolation by SEC.

Analytical size exclusion chromatography (SEC) for the mSerRS-mtRNA^{Ser(GCU)} complex. The mSerRS dimer runs as a single peak (blue line), which elutes at ca. 12.4 ml. The addition of mtRNA^{Ser(GCU)} at a 1.5-fold molar excess over mSerRS results in a shift of ca. 0.2 ml to an elution volume of 12.2 ml (orange line). The same shift is observed for the addition of a 5-fold molar excess (red line).

To further test the stoichiometry of the human mSerRS-mtRNA^{Ser(GCU)} complex, we used mass photometry, a technique which enables the accurate mass measurement of single molecules and complexes in solution and at low concentrations [Young et al, Science, 2018]. As shown in the figure below, the addition of a 5-fold excess of mtRNA^{Ser(GCU)} (125 nM) to mSerRS (25 nM) results in the formation of a single peak shifted by 20 kDa relative to the free mSerRS dimer, corresponding to a single mtRNA^{Ser(GCU)} (MW = 20.2 kDa). No additional peak is observed at higher MW, which would result from the formation of a 2:2 complex. Thus, mass photometry supports a 2:1 ratio of mSerRS to mtRNA^{Ser} at these concentrations. We included the new methods and data in the revised manuscript (line 383, line 92, and new Supplementary Fig. 7d).

Mass photometry suggests that one mtRNA^{Ser(GCU)} binds to the mSerRS dimer. 25 nM mSerRS were mixed with 125 nM mtRNA^{Ser(GCU)}. Complex formation resulted in a mass-shift of 20 kDa, corresponding to a single tRNA bound to the mSerRS dimer.

Why our cryo-EM and mass photometry results differ from those reported by Shimada et al [Shimada et al, JBC, 2001] we do not know. Possible reasons are the relatively high enzyme and tRNA concentrations used by Shimada et al (440 nM mSerRS and up to 5 μ M mtRNA^{Ser(GCU)}) and the use of EMSA to determine complex formation, which is known to stabilize complexes by caging effects. It may also be important to note that Shimada et al use the bovine system, raising the possibility of species-specific differences in the stability of 2:2 complexes.

Notably, structural and biochemical information available from non-mitochondrial SerRSs likewise suggest that the SerRS dimer preferably binds only one tRNA [Yaremchuk et al, FEBS, 1992; Biou et al, Science, 1994; Cusack et al, EMBO J, 1996], which led Biou et al to suggest that “this stoichiometry might have a particular biological significance” [Biou et al, Science, 1994]. This is consistent also with studies which suggested that the two tRNA^{Ser} binding sites of SerRSs are not equivalent [Krauss et al, Eur. J. Biochem., 1975; Rigler et al, Eur. J. Biochem., 1976; Wang et al, NAR, 2015]. The reasons for this non-equivalence are not known – even for canonical mSerRSs – and we agree with the reviewer that it would be very interesting to understand its molecular basis in mSerRS and whether it plays a functional role. However, the quality of the EM density in the corresponding regions of the model (with the second helical arm not defined in the density) is not sufficiently high to provide a detailed mechanistic explanation. In this context, it may also be important to keep in mind that we have no empirical information regarding the absolute or even relative concentrations of mt-aaRSs and mtRNAs in mammalian mitochondria, which would be important to understand the biological relevance of a 2:1 vs 2:2 stoichiometry. We therefore chose to not include a (necessarily speculative) discussion of the mechanistic basis or biological relevance for this asymmetry, especially as it is not the focus of this manuscript. We hope that the reviewer agrees with our decision.

A brief discussion of this point was included in the revised manuscript, together with the new mass photometry results (from line 92 and Supplementary Fig. 7d).

Regarding the active site titrations: This is a standard assay that is performed in our lab to determine the ‘number of active sites’ in our protein preparations. It monitors the formation of the seryl-adenylate and is not performed in the presence of tRNA. As described from line 357 onward, it does not provide information that is immediately relevant to the main arguments of the manuscript and was therefore not included. Below, we provide the active site titration for two of our mSerRS preparations, which shows that about >80% of the purified mSerRSs contain catalytically competent active sites.

Active site titration experiments to determine the number of catalytically competent active sites for two preparations of mSerRS(wt) and mSerRS(R146A). Reactions contained either 5 μ M (orange) or 10 μ M (blue) of purified human mSerRS, 20 mM L-serine, 22 nM [γ -³²P]-ATP, in assay buffer (100 mM HEPES pH 7.5, 20 mM KCl, 10 mM MgCl₂, 2 mM DTT, and 2 mg/mL pyrophosphatase) and were carried out at room temperature.

- The resolution and quality of the cryo-EM reconstructions are disappointing given modern standards. Authors should investigate if for example Bayesian polishing combined with refinement of global aberrations (divided for beam/image-shift groups from the Legion collection) can improve the resolution of the reconstructions especially since the ms describes and delineates important molecular aspects of tRNA structure and tRNA/protein interactions. We recognize that a higher resolution structure would be more satisfying, but the attainable resolution was limited by numerous factors: the targets are small in size, do not have symmetry, exhibit significant structural flexibility, and the tilting strategy required to overcome the preferred orientation issues made ice thicker, increased beam-induced motion, and generally increased the B-factors associated with data processing. We have used all available processing strategies to maximize the resolution of our structures and are confident in the models derived from our cryo-EM reconstructions.

We have implemented CTF refinement in our processing scheme for mSerRS-mtRNA^{Ser(GCU)}, see Methods section “Human mSerRS-mtRNA^{Ser(GCU)} cryo-EM data processing” and Supplemental Figure 5. Indeed, CTF refinement (including global aberrations and with particles grouped by image-shift) improved the resolution of the reconstruction. We attempted Bayesian polishing on this dataset as well, however it did not improve the reconstruction and was therefore not included in the manuscript. We now state this in the text (line 480).

In our efforts to generate a reconstruction for mSerRS-mtRNA^{Ser(GCU)-TL}, we implemented CTF refinement and Bayesian polishing in a separate processing scheme. That processing scheme did not yield a higher resolution reconstruction than the published method. We now explicitly state this in the text (line 497).

Minor comments:

5. Suppl. Table 1: R3 implies a hexagonal setting in MX so stating both a and b lengths and unit cell angles is superfluous. Version of Phenix used should be stated both in the table and in the methods text. Should be stated/noted that it is the average B-factor and please separate protein and ligand. For Ramachandran, rotamer and clash score statistics it should be stated if it is according to Molprobity or some other validation program.

The text and table were updated, and the requested information was included.

6. Suppl. Table 2: Typo for the unit in electron exposure. Version of Phenix used should be stated both in the table and in the methods text. It should be stated/noted that it is the average B-factor and please separate protein and tRNA. For Ramachandran, rotamer and clash score statistics it should be stated if it is according to Molprobity or some other validation program.

We thank the reviewer for catching the typo and missing information. We edited as follows:

- Suppl. Table 2: $e^- \text{Å}^{-2}$

- Suppl. Table 2: Phenix-1.10.1

- Methods lines 414, 510, 512: Phenix-1.10.1

- Suppl. Table 2: Average B factors were updated as suggested.

- Methods lines 512-513 were updated to "MolProbity was used to assess the quality of the final model and report validation statistics in Supplementary Table 2."

7. p. 9 "the" missing in parts in the descriptions relating to the N-helix and the C-tail.

The sentence was revised as suggested (line 200).

8. In the methods section, it is not detailed which pyrophosphatase is used (E. coli, yeast or..).

Yeast pyrophosphatase from Roche was used. The information was updated (lines 341, 360, 370).

9. p. 20, UltraAuFoil is from Quantifoil, EMS is a distributor.

We thank the reviewer for noting this. The information was updated (line 432).

10. p. 21 the initial model seems to have been generated twice, first from Warp-picked particles and second from LOG-picked particles? Please clarify.

We thank the reviewer for this suggestion to improve the clarity of the manuscript. An initial model was indeed generated from Warp-picked particles; however, it was not used in the processing scheme of the published reconstructions. The initial models that were used in the processing schemes were generated as described in the two following paragraphs of the methods. The sentence "Particle stacks from Warp..." on line 445 was edited to remove the confusion.

11. p. 21 "Iterative CTF and 3D auto refinement resulted". Here it is unclear if it is just the local defoci that are refined or also global aberrations? If also global aberrations, which of them and in which order?

First, one round of CTF refinement was performed on beam-tilt and trefoil only, followed by 3D auto-refinement. A second round of CTF refinement was run to refine per-particle defocus, astigmatism and B-factors, again followed by 3D auto-refinement. The sentence "Iterative CTF..." on line 478 was edited for clarification.

12. p. 22 For the TL-variant tRNA structure: It is unclear if local defoci and global aberrations (which of them and in which order) are refined either in CryoSPARC or after export to the RELION suite.

CTF refinement did not yield an improvement in resolution of mSerRS-mtRNA^{Ser(GCU)-TL}. This is now explicitly stated in the revised methods of the manuscript (line 497).

13. Suppl. Figures 5,6C. The resolution range chosen is strange. Surely there must be parts of the reconstruction that are better than 3.9-4 Å? A range of 3-6 or 3-5 Å would be more informative.

The range shown in Suppl Figures 5,6c was chosen to best illustrate the resolution range in the reconstruction. Please see below for figures with the suggested ranges:

14. RELION is used but not referenced.

We thank the reviewer for noting this. The reference was included (line 451).

15. The names of the software suite RELION should be capitalized throughout the ms. Please also capitalize XDS in ref. 50. Furthermore, Phenix is capitalized in the MX methods part but not in the cryo-EM part so please adjust for consistency (Phenix is preferred by the Phenix authors).

The manuscript was revised accordingly.

16. Suppl. Fig. 2B. Choose a thinner C- α representation than cartoon and choose a contrasting color scheme for overall structural comparison of the bovine and human mSerRS.

The figure was updated. We chose relatively weak colors for the body of bovine mSerRS to highlight the mSerRS-specific N-helix and C-tail.

17. MolProbity should have a capital P.

The typos were corrected.

Reviewer #3 (Remarks to the Author):

Review of Kuhle et al

This manuscript is bursting with exciting new data and insights. The authors have solved the structure of a complex between the human mitochondrial seryl-tRNA synthetase (mSerRS) and its cognate mitochondrial tRNA^{Ser}. The work is technically state of the art, and the conclusions are quite striking, documenting one of the more penetrating and interesting aspects of the structural biology of genetic coding to emerge in recent years.

Mitochondrial tRNAs have long been of considerable interest because they exhibit substantially more violations of the canonical “L-shaped” structure than do tRNAs from any other domain, in spite of the fact that the cognate aminoacyl-tRNA synthetases (aaRS), generally encoded in the nucleus, have almost exclusively canonically bacterial sequences and structures. Mitochondrial tRNAs, encoded by the mitochondrial genome, are often deficient in one or several of the features thought to be canonical, and which often include what are described as “identity” elements because they help define the specific recognition interfaces with cognate aaRS. Of the variant mitochondrial tRNAs, mitochondrial seryl-tRNA, mtRNA^{Ser}, deviates the most from canonical tRNAs, because it lacks the entire dihydrouridine stem loop, which contributes with the T ψ C loop to form the tertiary base pairs that form the core stabilizing the L shape. Kuhle et al have used X-ray crystallography to solve the structure of the mitochondrial seryl-tRNA synthetase, mSerRS, in the absence of its cognate mtRNA^{Ser}, and Cryo electron microscopy to solve its complexes with several mtRNA^{Ser} variants.

The structures reveal quite surprising consequences of the loss of the D-stem loop. The complex most closely resembles that of the corresponding bacterial complex in the interaction of the acceptor stem and T ψ C stem-loop, which at first glance appear quite homologous. The details, however, reveal quite significant changes in both the mtRNA^{Ser} structure and complementary differences in the mSerRS conformation. Most surprising is that in the absence of the canonical core tertiary base pairs, and perhaps in the absence of the long variable arm, the anticodon stem loop is untethered and exhibits a highly flexible structure that cannot be completely specified in any of the electron density maps, but which appears from the complex with a stabilized variant of mtRNA^{Ser} to form an acute angle that is aptly described as forming a “Y” shaped structure, rather than the canonical “L”-shape.

The extraordinary biological relevance of these differences is that nearly all of the canonical “Identity” elements that inform the specific recognition between aaRS and tRNA have been replaced by much more base-sequence tolerant secondary structural features. The most evident of these is that the T-stem is one base-pair longer than the canonical tRNA^{Ser}, which rotates and re-models the T-loop, allowing the two bases extruded in canonical tRNAs to form tertiary base triples with the (missing) D-loop to base pair instead with the 3' antiparallel strand of the T-loop. The long α -helical N-terminal helical coiled-coil domain curves more tightly around this re-configured T-loop, substantially

increasing the surface area buried between that domain and the minor groove of the elongated T-stem. That the essential contacts in the complex involve only the minihelix is remarkably consistent with Professor Schimmel's earlier suggestion that cognate interactions might have developed first between tRNA-like minihelices before the advent of the D- and anticodon-stem loops (Schimmel, et al., Proc. Nat. Acad. Sci. USA, 1993, 90, 8763-8768).

Acceptor stem interactions are more canonical, but even these are altered to increase the base sequence independence of the cognate interaction. The mSerRS motif 2 loop is shortened, precluding interactions that confer sequence-preferences in positions 2:71 and 3:70 by the bacterial SerRS. Those interactions are replaced instead by backbone amide hydrogen bonds. In this respect, as with the T stem-loop interaction, the changes tend to eliminate base sequence dependence of specific recognition.

These details and others are outlined clearly and without ambiguity, anticipating and answering adequately all the questions I might have raised from the structure and making the manuscript a joy to read. Moreover, the methods section describes a number of technically advantageous high-throughput modifications of routinely used methods as well as for synthesis of Ser-5' sulfamoyl adenylate.

We thank the reviewer for her/his positive feedback and the constructive comments and suggestions.

The only comment I might add is that the new structural details also have rather profound evolutionary implications, not only for the (relatively) recent evolutionary divergence of mitochondrial translation machinery, which the authors describe, but also in an opposite sense, for the early ancestry of quite early aaRS•tRNA cognate pairs. Acceptor-stem interactions with the motif 2 loop illustrate key elements of what differentiate Class II aaRS tRNA substrates from those for Class I tRNAs, namely the fully extended 3'DCCA. Further, dependence on primary sequence tolerant mechanisms such as structural shape rather than base sequence recognition were also probably key to the early formation of synthetase•tRNA cognate pairs (C. W. Carter, Jr and P. R. Wills, NAR, 2018, 46, 9667–9683; C. W. Carter, Jr and P. R. Wills, IUBMB Life, 2019, 71, 1088–1098).

We thank the reviewer for raising this interesting point, which we very much agree with. As the reviewer points out, our results have profound evolutionary implications far beyond the emergence of the animal mitochondrial translation machinery itself, which is why we think that they warrant a deeper investigation along the lines the reviewer lays out. We therefore tried to maintain the focus of this present manuscript on the molecular mechanisms of mitochondrial aaRS-tRNA interactions and their immediate implications for the evolutionary divergence of the animal mitochondrial translation machinery. We hope the reviewer agrees with our decision.

Minor criticisms:

A trivial, but important source of confusion occurs on line 74 of page 4. The α/β terminology is at variance with long-standing description of α/β proteins as those such as TIM barrel and Rossmann dinucleotide binding proteins, which are dominated by parallel β -strands, whereas Class II aaRS are primarily antiparallel β structures in which the alpha helices are segregated from the beta strands. Thus, this is a confusing nomenclature, violating a classification introduced by Levitt and Chothia in 1976, and which has remained in continuous use ever since. According to those authors, the segregation of strand and antiparallel strand structures in Class II aaRS is termed $\alpha+\beta$.

Thank you for pointing this out. The information was updated (line 73).

On line 111, page 6, the authors might remind readers that even canonical SerRS•tRNA^{Ser} complexes exhibit no specific recognition of the anticodon stem-loop because there are six serine codons two of which have a different middle base from the other four.

Thank you for this suggestion. Since we had difficulty inserting this information in line 111, we instead included it in line 91.

Reviewers' Comments:

Reviewer #1:

Remarks to the Author:

The authors have addressed my concerns satisfactorily.

Reviewer #2:

Remarks to the Author:

I think the revised MS and rebuttal adequately addressed the concerns raised. One remaining issue is the version of Phenix (1.10.1) the authors claims to have used. The 1.10.1 version was obsoleted in October 2016 and is not useful for cryo-EM refinement. Current release version is 1.20.1. Maybe a typo/misunderstanding?

Point-by-point response to the reviewer's comments

Reviewer #1 (Remarks to the Author):

The authors have addressed my concerns satisfactorily.

Reviewer #2 (Remarks to the Author):

I think the revised MS and rebuttal adequately addressed the concerns raised. One remaining issue is the version of Phenix (1.10.1) the authors claims to have used. The 1.10.1 version was obsoleted in October 2016 and is not useful for cryo-EM refinement. Current release version is 1.20.1. Maybe a typo/misunderstanding?

We thank the reviewer for catching this mistake. We indeed used the latest version (1.20.1). The information was updated.